# Inhalable Mannosylated Rifampicin–Curcumin Co-Loaded Nanomicelles with Enhanced In Vitro Antimicrobial Efficacy for an Optimized Pulmonary Tuberculosis Therapy

**DOI:** 10.3390/pharmaceutics14050959

**Published:** 2022-04-28

**Authors:** Juan M. Galdopórpora, Camila Martinena, Ezequiel Bernabeu, Jennifer Riedel, Lucia Palmas, Ines Castangia, Maria Letizia Manca, Mariana Garcés, Juan Lázaro-Martinez, Maria Jimena Salgueiro, Pablo Evelson, Nancy Liliana Tateosian, Diego Andres Chiappetta, Marcela Analia Moretton

**Affiliations:** 1Facultad de Farmacia y Bioquímica, Universidad de Buenos Aires, Buenos Aires 1113, Argentina; galdo.juan89@gmail.com (J.M.G.); eze_bernabeu@yahoo.com.ar (E.B.); jenn.driedel@gmail.com (J.R.); diegochiappetta@yahoo.com.ar (D.A.C.); 2Facultad de Ciencias Exactas y Naturales, Instituto de Química Biológica de la Facultad de Ciencias Exactas y Naturales (IQUIBICEN), CONICET, Universidad de Buenos Aires, Ciudad Universitaria, Buenos Aires 1113, Argentina; cami.b.m254@gmail.com (C.M.); nantateosian@gmail.com (N.L.T.); 3Consejo Nacional de Investigaciones Científicas y Técnicas (CONICET), Buenos Aires 1425, Argentina; mjsalguei@gmail.com; 4Instituto de Tecnología Farmacéutica y Biofarmacia (InTecFyB), Universidad de Buenos Aires, Buenos Aires 1113, Argentina; 5Department of Scienze della Vita e dell’Ambiente, University of Cagliari, 09124 Cagliari, Italy; luciapalmas@yahoo.it (L.P.); ines.castangia@unica.it (I.C.); 6Departamento de Química Analítica y Fisicoquímica, Cátedra de Química General e Inorgánica, Facultad de Farmacia y Bioquímica, Instituto de Bioquímica y Medicina Molecular (IBIMOL), CONICET, Universidad de Buenos Aires, Buenos Aires 1113, Argentina; msgarces87@gmail.com (M.G.); pevelson@gmail.com (P.E.); 7Departamento de Química Orgánica, Facultad de Farmacia y Bioquímica, Instituto de Química y Metabolismo del Fármaco (IQUIMEFA), CONICET, Universidad de Buenos Aires, Buenos Aires 1113, Argentina; jmlazaromartinez@gmail.com

**Keywords:** polymeric micelles, Soluplus^®^, rifampicin, curcumin, tuberculosis, inhalable nanoformulation, active drug targeting, *Mycobacterium tuberculosis*

## Abstract

Among respiratory infections, tuberculosis was the second deadliest infectious disease in 2020 behind COVID-19. Inhalable nanocarriers offer the possibility of actively targeting anti-tuberculosis drugs to the lungs, especially to alveolar macrophages (cellular reservoirs of the *Mycobacterium tuberculosis*). Our strategy was based on the development of a mannose-decorated micellar nanoformulation based in Soluplus^®^ to co-encapsulate rifampicin and curcumin. The former is one of the most effective anti-tuberculosis first-line drugs, while curcumin has demonstrated potential anti-mycobacterial properties. Mannose-coated rifampicin (10 mg/mL)–curcumin (5 mg/mL)-loaded polymeric micelles (10% *w*/*v*) demonstrated excellent colloidal properties with micellar size ~108 ± 1 nm after freeze-drying, and they remain stable under dilution in simulated interstitial lung fluid. Drug-loaded polymeric micelles were suitable for drug delivery to the deep lung with lung accumulation, according to the in vitro nebulization studies and the in vivo biodistribution assays of radiolabeled (99mTc) polymeric micelles, respectively. Hence, the nanoformulation did not exhibit hemolytic potential. Interestingly, the addition of mannose significantly improved (5.2-fold) the microbicidal efficacy against *Mycobacterium tuberculosis* H37Rv of the drug-co-loaded systems in comparison with their counterpart mannose-free polymeric micelles. Thus, this novel inhaled nanoformulation has demonstrated its potential for active drug delivery in pulmonary tuberculosis therapy.

## 1. Introduction

Pulmonary drug delivery or “orally inhaled therapy” has been extensively tested in recent years due to the advantages of this administration route and the wide versatility on novel inhaled dosage forms [1]. The former includes (1) drug administration for both local (i.e., tuberculosis (TB), SARS coronavirus disease, chronic obstructive pulmonary disease and cystic fibrosis, pneumonia, asthma, lung cancer), and systemic diseases such as diabetes (Afrezza^®^) and migraine (Migranal^®^) [2,3], associated with a significant reduction of the systemic side-effects and hepatic first-pass metabolism; (2) the rapid onset of action; and (3) the possibility to specifically transport the drug in a precise site by the addition of sugar residues, polysaccharides, antibodies, peptides and small molecules capable of enhancing cellular uptake [4,5] 

Different inhalable dosage forms such as solutions for nebulization [6] and dry powders for aerosolization [7] have been developed in the last decades aiming at optimizing the pulmonary therapy. One of the main drawbacks for inhalable dosage forms are the anatomical and physiological barriers (i.e., mucocilliary clearance, proteolytic enzymes and phagocytic cells) of the lungs [8]. In this context nanotechnology provides a feasible platform for the development of novel respirable drug delivery systems. For instance, fluticasone-loaded liposomes and cyclodextrins complexes [6]; chitosan/thiolated chitosan nanoparticles conjugated with hyaluronic acid loaded with isoniazid [4]; and antiviral-loaded polymeric nanoparticles for SARS-CoV-2 inhibition [9] highlight the nanotechnological platforms recently investigated for the treatment of lung diseases by means of inhaled therapy.

Among respiratory infections, TB is a chronic infectious disease that affects people globally. This disease is caused by *Mycobacterium tuberculosis* (Mtb). As reported by World Health Organization in 2021, TB was the second deadliest infectious disease in 2020 behind the COVID-19 global pandemic. Furthermore, 10 million cases were reported, and 1.5 million patients died because of TB infection [10].

Currently, TB is treated with a combined therapy that includes “first-line” anti-TB drugs as rifampicin (RIF), isoniazid, pyrazinamide and ethambutol for 6 months (short-term treatment). These “first-line” drugs are administered orally [11]. Nevertheless, side effects such as peripheral neurotoxicity, liver toxicity and renal toxicity affect both patient life-quality and adherence to the treatment. Consequently, these aspects lead to treatment failure and an increase of multi-drug resistant of TB strains [11].

In particular, RIF is the most effective first-line anti-TB drug, and it is currently classified as a borderline Class II drug by the Biopharmaceutics Classification System because of its low pH-dependent water solubility (2.56 mg/mL 25 °C, pH: 5.0) and low intestinal permeability [11]. Additionally, due to its low water solubility and poor chemical stability, there is a lack of liquid or aqueous RIF pharmaceutical dosage forms. Furthermore, simulation models of RIF pharmacokinetics/pharmacodynamics in lungs showed that the standard 600 mg RIF oral-dose could not prevent the development of drug resistance due to the low pulmonary RIF concentrations [12].

In this framework, novel inhalable RIF dosage forms are required in order to optimize TB therapy and enhance patient adherence. Moreover, previous studies have shown the potential of inhalable anti-TB drugs as a promising alternative to the current oral TB treatment [13]. Particularly, it is well known that Mtb is mainly hosted by alveolar macrophages (AMs), which exhibit in their surface mannose receptors (C-type lectin, CD206) involved in Mtb phagocytosis and responsible for the immune response in TB infections [14]. Patients with pulmonary TB showed an enhanced expression of the mannose receptor in both lung and pleural tissues [15]. Then, this receptor appears as an excellent target for an active drug delivery of anti-TB drugs to the Mtb cellular reservoirs.

Curcumin (CUR) is a polyphenol compound obtained from Curcuma longa root currently used as an oral nutraceutical [16] and as an anti-inflammatory agent in Ayurveda medicine. Nevertheless, CUR exhibits low aqueous solubility [17], which hampers the development of liquid CUR dosage forms.

Recently, the potential anti-mycobacterial properties of CUR have been investigated, since it is known that it can modulate the host immune response [18]. Furthermore, in a previous study, the association of curcumin and rifampicin by means of a nanotechnological system based on polyethylene sebacate nanoparticles has been formulated and tested for oral administration, confirming the ability of these systems to improve the in vitro mycobacterial clearance [19].

In a previous study, a respirable RIF-loaded nanocarrier based in Soluplus^®^ polymeric micelles demonstrated to have a significantly improved in vitro microbicidal efficacy in comparison with the RIF solution [20]. Soluplus^®^ is a graft copolymer of poly (vinyl caprolactam)-poly (vinyl acetate)-poly (ethylene glycol) (PEG) that showed good performance solubilizing poorly soluble drugs [15,16,17] and high micellar stability (due to its low CMC value). Hence, the present investigation was aimed at expanding the potential of this inhalable nanotechnological platform for an active drug targeting to the Mtb-infected macrophages. Given that, Soluplus^®^ micelles decorated on the surface with mannose residues co-loading rifampicin and curcumin have been formulated and deeply characterized in terms of size, size distribution and morphology. Moreover, the in vitro nebulization aptitude of formulations and their aerodynamic diameter were evaluated along with their hemolytic potential. Finally, the microbicidal effectiveness of formulations and the micellar accumulation in the lungs has been investigated in vitro by using THP-1 infected (Mtb H37Rv) macrophages and in vivo by using Wistar rats, respectively.

## 2. Materials and Methods

### 2.1. Materials

Polyvinyl caprolactam–polyvinylacetate–PEG 6000 (Soluplus^®^, average MW~120,000 g/mol) was a kind gift of BASF (Buenos Aires, CABA, Argentina). Rifampicin (RIF) and dried bovine gelatin (Bloom 125, type B, average MW ~125,000 g/mol) were purchased from Parafarm^®^ (CABA, Argentina). Curcumin (CUR), bovine serum albumin (BSA), concanavalin A (Con A, from Canavalia ensiformis, Jack Bean Type VI), mannose and mucin II (from porcine stomach) were acquired from Sigma-Aldrich (Buenos Aires, CABA, Argentina). All solvents were of grade and used following the manufacturer’s instructions. 

### 2.2. Gelatin/Mannose Formulation and Characterization

Gelatin–mannose (Gel(man)) was prepared using dried bovine gelatin and mannose as previously described with slight modifications [21]. Dried bovine gelatin (750 mg) was dispersed in 15 mL of distilled water at 45 °C under magnetic stirring (50 RPM) for 30 min. Meanwhile, mannose (80 mg) was solubilized in 10 mL of AcH/AcNa buffer (pH 4.20) under magnetic stirring (50 RPM) for 30 min. Afterwards, gelatin dispersion and mannose acid solution were mixed under magnetic stirring (100 RPM) at room temperature (25 °C) for 72 h. Finally, the resulting dispersion was dialyzed against distilled water using dialysis membranes (Spectra/Por^®^3 Dialysis Membrane, molecular weight cut off = 3500 Da, nominal flat width 18 mm, Merck SA, Buenos Aires, Argentina) over 1 h (replacing the external medium every 15 min) at room temperature. The dialyzed samples were then freeze-dried for 48 h at 0.03 mbar, using a freeze-dryer FIC-L05 (FIC, Scientific Instrumental Manufacturing, Buenos Aires, Argentina), to improve the stability of the sample in storage.

Gel(man) has been deeply characterized by means of different methods. Diamond ATR-FTIR (attenuated total reflectance Fourier-transform infrared spectroscopy) spectra were acquired using a Nicolet iS50 Advanced Spectrometer (Thermo Scientific, Waltham, MA, USA) with 64 scans and a resolution of 4 cm^−1^. Proton Nuclear Magnetic Resonance (^1^H-NMR) experiments were acquired with a Bruker Avance-III HD spectrometer equipped with a 14.1 T narrow bore magnet operating at Larmor frequencies of 600.09 MHz. Chemical shifts for ^1^H (in ppm) are relative to Si(CH_3_)_4_. Gel(man) has been used to decorate Soluplus^®^ micelles, and the presence of mannose residues on their surface/corona was evaluated using Concanavalin A (Con A) [22]. Briefly, Soluplus^®^ was dispersed in phosphate buffer (pH 7.2) in order to obtain a final concentration of 10% *w*/*v*. Then, 100 mg of Gel(man) were added to the Soluplus^®^ dispersion under magnetic stirring (50 RPM) until complete Gel(man) dispersion. Control Gel(man)-free Soluplus^®^ micelles (10% *w*/*v*) were used for comparison. In a second step, bovine serum albumin (BSA, 90 mg) was added to the micellar dispersions with and without Gel(man) and Soluplus^®^ (man)-micelles (2 mL). Samples were magnetically stirred for 30 min at 25 °C and diluted (1/2) with a Con A (10 µM) phosphate buffer solution (pH: 7.2). Finally, the micellar systems were magnetically stirred (100 RPM) for 2 h at room temperature, and the micellar size and size distribution was investigated by dynamic light scattering (DLS, scattering angle of θ = 173° to the incident beam, Zetasizer Nano-ZSP, ZEN5600, Malvern Instruments, Malvern, UK) at 25 °C (n = 5 ± S.D.).

### 2.3. Measurement of the Critical Micellar Concentration (CMC)

The CMC at 25 °C of Soluplus^®^ and Soluplus^®^-Gel(man) (0.7% *w*/*v*) in water was assessed by dynamic light scattering (DLS), using a Zetasizer Nano-ZSP (ZEN5600, Malvern Instruments, Malvern, UK), at a scattering angle of Φ = 173° to the incident beam. Micellar aqueous dispersions were prepared to obtain a concentration range between 1 × 10^−6^ and 1% *w*/*v*. Samples were equilibrated for 24 h at 25 °C before their use. The CMC was graphically determined by plotting the derived count rate as a function of the polymer concentration (% *w*/*v*) [22].

### 2.4. Micellar Preparation and Drug Encapsulation 

Soluplus^®^ (1 g) was dispersed in distilled water (10 mL) under magnetic stirring (50 RPM, 1 h) at room temperature (25 °C) to obtain “drug-free-micelles” and stored overnight before the use. Soluplus^®^ micelles decorated with Gel(man) so called “drug-free-(man)micelles”, were prepared as well. The graft-copolymer (Soluplus^®^, 1 g) was dispersed in water (10 mL), and the appropriate amount of Gel(man) (25, 50, 70, 100 mg) was added under magnetic stirring (50 RPM) until complete and homogeneous dispersion.

In order to encapsulate both drugs, RIF and CUR, the solvent diffusion method was used, as previously described [17,20]. Briefly, RIF (100 mg) and CUR (50 mg) were dissolved in 20 mL of acetone; the solution was sonicated for 5 min (Digital Ultrasonic Cleaner, PS-10A 50/60 Hz, Shenzhen, China, 25 °C) and then added drop by drop using a programmable syringe infusion pump (PC11UB, APEMA, Buenos Aires, Argentina) to the aqueous dispersion of Soluplus^®^ (10% *w*/*v*) at room temperature under constant magnetic stirring (50 RPM, 4 h) to let the complete evaporation of the organic solvent, as well. Afterwards, the volume of RIF–CUR co-loaded-micelles, so called “RIF–CUR-micelles”, was repristinated at 10 mL using distilled water and samples were filtered using 0.45 μm, acetate cellulose filters (Microclar, Buenos Aires, Argentina), aiming at removing the undissolved/unentrapped drugs.

Drug-loaded, micelles-decorated mannose, so-called “RIF–CUR-(man)micelles”, were prepared following the same procedure, with small modifications. In this case after the acetone evaporation, Gel(man) (25, 50, 70, 100 mg) was added under magnetic stirring at 25 °C until complete dispersion. Then, the procedure was performed as described above for the mannose-free, drug-loaded micelles.

Finally, drug-free and drug-loaded nanomicelles were freeze-dried for 24 h at 0.03 mbar using a freeze-dryer, FIC-L05 (FIC, Scientific Instrumental Manufacturing, Argentina), and stored in sealed amber glass vials until their use.

The concentrations of RIF and CUR in the samples were determined using a Shimadzu HPLC system (SIL-10A auto sampler, SCL-10A pump and SPD-10AV UV detector, Japan). All separations were achieved with an Atlantis-C18 reversed-phase column (4.6 mm × 150 mm, 3 μm particle size, Waters, Ireland) at a flow rate of 1 mL/min and at room temperature. The mobile phase consisted of acetonitrile:water (adjusted to pH 2.27 with orthophosphoric acid). For RIF measurement, the ratio of mobile phase was 54:44 (*v*/*v*), and the detection wavelength was 333 nm. The CUR detection wavelength was 425 nm, and the mobile phase was fixed at 44:56 (*v*/*v*). Twenty microliters of all the sample solutions were injected.

### 2.5. Measurement of Micellar Size and Morphological Characterization

Micellar size, size distribution and polydispersity index (PDI) of the nanoformulations was assessed at 25 °C by dynamic light scattering (DLS) using a Zetasizer Nano-ZSP (ZEN5600, Malvern Instruments, Malvern, UK) at a scattering angle of θ = 173° to the incident beam. Prior to the analysis, each sample was equilibrated at 25 °C. Finally, the results of hydrodynamic diameter (Dh) and PDI values were expressed as the average of five measurements ± S.D.

To assess the morphology of the RIF–CUR-loaded (10 mg/mL and 5 mg/mL) (man)micelles, a transmission electron microscopy (TEM) analysis was performed using a TEM apparatus (Philips CM-12 FEI Company, Eindhoven, The Netherlands). Freeze-dried samples were re-dispersed in distilled water (2 mL), and aliquots (5 μL) were negatively stained with 5 μL of uranyl acetate (2% *w*/*v*).

### 2.6. In Vitro Antioxidant Capacity Assays

#### 2.6.1. DPPH Colorimetric Assay

The antioxidant activities of the different formulations were evaluated on the basis of the scavenging activity of the stable 2,2-diphenyl-1-picrylhydrazyl free radical (DPPH•). Aliquots of methanolic solutions (1 mg/mL) of the different formulations were incubated with 3 mL of a methanolic solution of DPPH• (25 mg/L). After 10 min, the absorbance of the mixture was measured at 517 nm. A calibration curve was prepared using Trolox (vitamin E analogue) as standard, and results were expressed as nmol Trolox Eq/mg sample and DPPH inhibition percentage [23].

#### 2.6.2. ABTS Colorimetric Assay

Using 2,2-azino-bis-3-ethylbenzothiazoline-6-sulphonic acid, or ABTS, a radical cation can be generated. The ABTS is generated by reacting with a strong oxidizing (ABAP, 2 mM) agent with the ABTS salt (75 mM) for an hour at 45 °C. The reduction of the blue-green ABTS radical by hydrogen-donating antioxidants absorbs light at 734 nm in phosphate buffer. Aliquots of aqueous dispersions (1 mg/mL) of the different formulations were incubated with 3 mL of ABTS• solution. After 4 min, the absorbance of the mixture was measured at 734 nm. A calibration curve was prepared using Trolox (vitamin E analogue) as standard, and results were expressed as nmol Trolox Eq/mg sample and ABTS inhibition percentage [24].

### 2.7. In Vitro Nebulization Studies 

The in vitro deposition of RIF and CUR (alone or in combination) was assessed using the next-generation impactor (NGI, Eur. Ph 7.2, Copley Scientific Ltd., Nottingham, UK) connected with the PariSX^®^ air jet nebulizer and the ParyBoySX^®^ compressor. Freeze-dried micelles (Soluplus^®^ 10% *w*/*v*, Gel(man) 0.7%, RIF 10 mg/mL, CUR 5 mg/mL) were re-dispersed in 2 mL of water, manually shaken and placed in the nebulizer. The aerosolization process was performed for about 15 min directly into the throat of the New Generation Impactor (NGI). The corresponding dispersions in water at the same concentration of RIF, CUR or both were also nebulized and used as references. The amount of RIF and/or CUR deposited in each stage of the NGI along with the undelivered drug were collected by using methanol and analysed by using a spectrophotometer. Aerodynamic diameter (MMAD) and geometric standard deviation (GSD) values were calculated excluding the amount of drugs deposited in the induction port and were both extrapolated from the graph obtained by plotting the cumulative amount of particles with a diameter lower than the stated size of each stage as a percentage of each recovered drug versus the cut-off diameter, according to the European Pharmacopeia [25,26]. The total mass output (TMO%), the Fine Particle Dose (FPD), and the Fine Particle Fraction (FPF), which represent the percentage of drug recovered in the NGI versus the amount initially placed in the nebulizer, the amount of drug contained in droplets of size less than 5 μm, and the percentage of droplets with size less than 5 μm, respectively, were measured as previously reported [27].

### 2.8. Stability of the Micellar Systems in Simulated Interstitial Lung Fluid

The in vitro physicochemical stability of the inhalable RIF–CUR-micelles with and without mannose residues was investigated after dilution. Simulated Interstitial Lung Fluid (SILF) was obtained by dissolving 9.85 mg MgCl_2_, 603.2 mg NaCl, 40.4 mg KCl, 43.6 mg Na_2_HPO_4_, 259.6 mg NaHCO_3_, 8.3 mg Na_2_SO_4_, 35.9 mg CaCl_2_, 60.7 mg C_2_H_3_NaO_2_ and 11.5 mg of sodium citrate in 100 mL of distilled water (pH 7.40) [28]. Freeze-dried RIF–CUR (10 mg/mL and 5 mg/mL)-(man)micelles (10% *w*/*v*) and freeze-dried RIF–CUR-(10 mg/mL and 5 mg/mL)-micelles (10% *w*/*v*) were re-dispersed in distilled water (2 mL) and then diluted (1/50) in SILF. Micellar size distribution and PDI were assessed at 0, 1, 2, 3 and 24 h after dilution at 37 °C by dynamic light scattering using a Zetasizer Nano-ZSP, ZEN5600 (Malvern Instruments, Malvern, UK) at a scattering angle of θ = 173° to the incident beam (n = 5 ± S.D.).

### 2.9. In Vitro Drug Release Study

The CUR and RIF cumulative in vitro release from the nanomicelles with (0.7% *w*/*v*) and without Gel(man) was studied by using dialysis method in SILF (pH 7.4) with the addition of ethanol (30% *v*/*v*) [17] as external medium. Lyophilized micellar systems were redispersed in distilled water and further diluted with distilled water in order to achieve concentrations of 1 mg/mL for RIF and 0.5 mg/mL for CUR. Later, 0.5 mL aliquots (500 µg of RIF and 250 µg of CUR) were placed into dialysis membranes (3500 Da, Spectra/Por^®^3 Dialysis Membrane, molecular weight cut off = 3500, nominal flat width 18 mm, Waltham, MA, USA), which were then placed inside Falcon^®^ conical tubes containing the external medium (12 mL). The system was incubated at 37 °C for 1, 2, 3, 4, 6, 8 and 24 h by inversion using a sample rotator (Mini Labroller LabNet rotator, St. Louis, MO, USA, 40 RPM). For every timepoint, the external medium was completely replaced with fresh and pre-heated medium. The amount of RIF and CUR was determined by RP-HPLC-UV-Vis, as previously described (Section 2.4). Assays were performed in triplicate, and results are expressed as average ± S.D.

### 2.10. In Vitro Hemolytic Assay

Hemolytic cytotoxic effects of the prepared micelles were determined in vitro. Fresh blood from control rats was collected in hemolysis tubes containing 3.2% of sodium citrate (1:9). Red blood cells (RBCs) were separated by centrifugation at 3500 RPM for 10 min and then acquired and washed with saline solution (NaCl 0.9% *w*/*v*). Afterwards, samples were centrifuged 3 times (3500 RPM, 10 min, MiniSpin^®^ plus™, Eppendorf, Hamburg, Germany), and the supernatant was discarded while the pellets were diluted with saline solution to reach the final concentration of 10% *w*/*v* to be used to evaluate the hemolytic activity of the formulations. The hemolytic assay was performed as previously described [29], with slight modifications. Saline solution was used as the negative control (0% lysis), and distilled water as the positive control (100% lysis), employing a 1/2 dilution with the RBCs suspension.

Freeze-dried samples (RIF, RIF–CUR-(man)micelles and drug-free-(man)micelles) were re-dispersed in distilled water (2 mL) and then diluted (1/2) with the RBCs suspension. The final RIF concentrations were 1, 5, 10 and 20 µg/mL. They were incubated at 37 °C for 3 h by inversion using a sample rotator (Mini Labroller LabNet rotator, USA, 40 RPM); then, samples were centrifuged for 10 min at 3500 RPM (800 RCF) (MiniSpin^®^ plus™, Eppendorf, Germany). The supernatant was processed, and hemoglobin released was measured at 541 nm by means of visible spectroscopy (8452A Diode Array Spectrophotometer, Hewlett Packard, Palo Alto, CA, USA). Later, the hemolysis percentage was calculated using the following equation:Hemolysis (%) = (Abs sample − Abs negative)/(Abs positive − Abs negative) × 100
where Abs negative and Abs positive correspond to the absorbance of the erythrocytes treated with saline solution and distilled water, respectively. Animal experiments and animal care were approved by the Animal Care Committee of School of Pharmacy of the University of Buenos Aires (REDEC-2021-2792-E-UBA-DCT_FFYB). Assays were performed in triplicate and expressed as mean ± S.D.

### 2.11. In Vivo Micellar Lung Accumulation Assay

#### 2.11.1. Radiolabeling Procedure of Soluplus^®^ (Man)Micelles

Freeze-dried micelles were re-dispersed in distilled water (2 mL), and then they were radiolabeled using stannous chloride (SnCl2; analytical grade Merck, Germany) as the reducing agent, as previously reported, with slight modifications [20]. Briefly, 1 mL of the aqueous micellar dispersion (3 mg/mL) was prepared and then mixed with 25 μL of and acidic solution (pH = 3.0, acidified with HCl 0.1 N) of SnCl2 (1 mg/mL) and 300 μL of freshly prepared sodium pertechnetate (Na99mTcO4; 1.5 mCi) eluted from an a99Mo/99mTc generator (Laboratorios Bacon SAIC, Villa Martelli, Argentina). The pH of the final dispersion was adjusted up to 7.0 using NaOH (0.1% *w*/*v* aqueous solution), and it was maintained for 60 min at room temperature in a closed chamber aiming at reducing the exposition of the samples to air.

#### 2.11.2. Radiolabeling Efficiency and Stability

The obtained 99mTc-(man)micelles were assayed for both labeling efficiency and radiochemical purity. Ascending chromatography was performed to detect free TcO4−, using Instant Thin Layer Chromatography-Silica Gel (ITLC-SG, Varian, Palo Alto, CA, USA) as the stationary phase and acetone as the mobile phase. The same stationary phase (ITLC-SG) but a different mobile phase (mixture of pyridine:acetic acid:water, 3:5:1.5) were used to detect 99mTc-radiocolloids and hydrolysate. The same procedure was used to test the effectiveness and stability of the after-incubation of 99mTc-(man)micelles in vitro using a saline solution at room temperature and in rat plasma at 37 °C for 24 h.

#### 2.11.3. Biodistribution Studies: Ex Vivo and Scintigraphic Procedures

Experimental tests with animals were completed according to the experimental protocol approved by the ethical committee of the School of Pharmacy and Biochemistry, University of Buenos Aires (REDEC-2021-2792-E-UBA-DCT_FFYB). Female Sprague-Dawley rats (6 weeks of age; 200 ± 10 g) were provided by the animal house (School of Pharmacy and Biochemistry, University of Buenos Aires). Animals were allowed to acclimate for at least 48 h before the experiment, housed in stainless steel cages with free access to both food and water and 12 h of light/dark cycles. Before the experiment, animals were anesthetized using ketamine–xilazine (90 mg/kg–5 mg/kg), and the radiolabeled micelles were administered by means of a surgical puncture tracheotomy (~0.3 mg micelles per rat, ~50 μCi); then, the animals were maintained at a supine position (45°) and with heads up during the first hour of the biodistribution assay. After 1 and 24 h of the administration procedure, the rats in each group (n = 5) were anesthetized using isofluorane 2% (Cassara, Buenos Aires, Argentina) and subjected to a static scintigraphy to detect the distribution of the radiolabeled (man)micelles by a view gamma camera, which was equipped with a high-resolution parallel hole’s collimator. Images were then processed with OHIO NUCLEAR, software IM512P (Alfanuclear, Buenos Aires, Argentina).

At the end of the experiment, rats were euthanized; blood samples were taken, and the organs of interest (liver, spleen, lungs, kidneys, heart, intestine and stomach) were removed, washed and weighted. The radioactivity in each organ was measured in a calibrated well type gamma counter (Alfanuclear, Argentina), and the results have been expressed as % of the injected dose of tissue, where decay correction was considered for the calculation.

### 2.12. Microbicidal Efficacy against Mycobacterium tuberculosis

#### 2.12.1. Bacterial Growth Conditions

*Mycobacterium tuberculosis* H37Rv (Mtb H37Rv) was grown in Middlebrook 7H9 broth or on 7H10 agar with 0.5% Tween 20, 0.2% glycerol, and albumin–dextrose–catalase–oleic acid supplement. Cultures were harvested at an exponential growing phase at 37 °C. To disaggregate clumps, mycobacteria were sonicated at 2.5 W output for 4 min (Elma d-7700 Singentrans sonic), then centrifuged for 10 min at 300× *g*, and the supernatant was diluted in PBS. Finally, the OD at 600 nm was determined. The bacterial growth of Mtb H37Rv and any experiment involving the pathogenic strain were performed in BSL3 security cabinets at the ANLIS-Malbran Institute, Buenos Aires, Argentina.

#### 2.12.2. Cell Culture and Infection

Human monocyte cell line (THP-1, ATCC TIB-202) was purchased from the American Type Culture Collection (Manassas, VA, USA). RPMI-1640 (Gibco, 22400–071) supplemented with 10% FBS (Gibco, 10437028), L-glutamine (2 mM; Sigma, G5792), penicillin-streptomycin and 2-mercaptoethanol (0.05 mM; Gibco) was used as the medium. Cells were cultured at 37 °C in a humidified atmosphere with 5% CO_2_. Macrophages were obtained by adding 10 ng/mL of phorbol-12-myristate-13-acetate (PMA; EMD Biosciences, La Jolla, CA, USA) for 48 h in 96-wells flat-bottom plates. After differentiation, THP-1 cells were infected with Mtb H37Rv (MOI 10) for 2 h. Then, cells were washed twice with warm RPMI and cultured in a complete medium without penicillin-streptomycin and in presence of (i) freeze-dried drug-free Soluplus^®^ (10% *w*/*v*) micelles, (ii) freeze-dried RIF-(10 mg/mL)-micelles (10% *w*/*v*) and (iii) freeze-dried RIF–CUR (10 mg/mL and 5 mg/mL)-micelles (10% *w*/*v*) in the presence and absence of Gel(man) 0.7% *w*/*v* for 48 h. The final drug concentrations investigated were: (i) RIF (5 µg/mL) and (ii) CUR (2.5 µg/mL). Freeze-dried micelles were re-dispersed in distilled water (1 mL) before their use.

#### 2.12.3. Colony-Forming Unit Assay

THP-1 differentiated into macrophages and infected with Mtb H37Rv were washed 2 times with (200 μL) of warm PBS and lysed with 0.05% Triton X-100 in PBS. The serial dilution of adherent cells lysates was made, and 10 μL aliquots were inoculated (in triplicate) on Middlebrook 7H10 agar plates supplemented with oleic acid–albumin–dextrose–catalase (OADC BD BBL Middelebrook cat 212351. 10% *v*/*v*). Plates were incubated for 3 weeks, and colonies were counted from dilutions yielding 10-100 visible colonies.

#### 2.12.4. Statistical Analysis

Statistical analysis was performed by one-way ANOVA test and Tukey’s multiple comparisons post-hoc test using GraphPadPrism version 6.01 for Windows (GraphPad Software, San Diego, CA, USA).

Statistical analysis concerning the colony-forming unit was performed by means of analysis of variance (ANOVA) and the post-hoc Tukey multiple comparisons test (*p* < 0.05).

## 3. Results and Discussion

### 3.1. Gelatin–Mannose Preparation and Characterization

One of the main goals of the present investigation was the development of a novel inhalable micellar formulation for an active drug targeting macrophages. In this way, the enhancement of the anti-TB therapy has been in the spot in recent years, mainly associated with the lack of patient adherence to the oral pharmacological treatment and the development of MDR-Mtb strains.

It has been found that the mannose receptor of the AMs is involved in the immune response after TB infection [14], and its over-expression has been reported in pulmonary TB patients [15]. Thus, this receptor can be considered an excellent target for an active drug delivery therapy to the cellular Mtb reservoirs in the lungs.

In this framework, we prepared mannose-modified gelatin in order to decorate the surface of the co-loaded (RIF and CUR) micellar nanoformulations. Gelatin is a natural biopolymer encoded as a GRAS material by the FDA [30]. Gelatin has been extensively employed in pharmaceutical, cosmetic and food industries and in drug/gene delivery systems since it is a biodegradable, biocompatible and low immunogenic polymer, with a great potential of chemical modifications to fine tune its biofunctional properties [31]. Further combining gelatin with other biomaterials has been explored as an attractive strategy to develop novel nano-drug delivery systems, such as poli(epsilon caprolactone)/gelatin based nanofibers [32]. Even more, the nanotechnological approach of employing mannosylated gelatin nanoparticles has been investigated as an attempt to modify the surface of nanoformulations for an improved anti-TB therapy employing linezolid [33]. 

Herein, we successfully developed mannose-modified gelatin as it could be observed by ATR-FTIR and ^1^H-NMR studies. Firstly, the ATR-FTIR spectrum for gelatin shows the characteristic peaks of the amide I (C=O stretching vibrations), amide II (N–H bending vibrations) and amide III (C–N stretching vibrations) at 1630, 1535 and 1241 cm^−1^, respectively (Figure 1A) [34]. Then, the ATR-FTIR spectra for gelatin and mannose-modified gelatin show an important difference in the stretching bands between 1200 and 1100 cm^−1^ with a new band at 1136 cm^−1^ associated to the C–O–C asymmetric stretching vibration frequency of mannose (Figure 1A,B) [35]. This situation is completely different from a physical mixture between gelatin with 5% mannose where the sum of the bands in the IR spectrum of the components is evident (Figure 1C). Although the nature of the interaction between both molecules cannot be known from the experiment, it is seen that the stretching of the C-O bonds typically found for mannose is present in the modified gelatin sample with a width and intensity different from the reference monosaccharide (Figure 1B–D).

Then, and in order to identify the nature of the interaction between mannose and gelatin, ^1^H-NMR experiments in deuterated water (D_2_O) were performed for the pure components and for the modified gelatin (Figure 2). 

It can be observed in the different ^1^H-NMR regions that mannose is present. Furthermore, the presence of the anomers (α and β) for the monosaccharide is observed at proton chemical shifts (δ^1^H) of 5.1 and 4.8, respectively. In addition, some other hydrogens can be assigned for the rest of the molecule for each anomer, as indicated in Figure 2 [36]. Although the spectrum for gelatin is complex due to the diverse amino acid composition, the aromatic region does not show resonance signals, associated with the formation of an imine or Schiff’s base (δ^1^H 7–9 ppm) [37], among the components of interest. The only signal present at δ^1^H~7.4 ppm is assigned to the hydrogens of the phenylalanine residues present in both gelatin and mannose-modified gelatin [38]. On the other hand, changes associated with proline residues close to δ^1^H~2 ppm are evidenced, which could indicate a molecular interaction between gelatin and mannose that also affects glutamic acid residues at δ^1^H~2.3 ppm. In a comparative way, it can be observed how the signals for the arginine residues (δ^1^H = 1.5–1.8 ppm) do not undergo changes in the presence of mannose. Additionally, the observed changes in the NMR spectrum may be the result of the changes or rearrangement in the gelatin structure after the interaction with mannose.

To complement the analysis, it was possible to estimate the mannose ratio in the modified gelatin. For the quantitative NMR analysis, the signals for the anomeric mannose hydrogens were considered, and they were related to the hydrogens for the methylene protons of the lysine residues next to the ε-amino group at δ^1^H = 2.9 ppm (-CH_2_-NH_2_). In this way, and considering that the average content of *Lys* in gelatin samples is 4% [39], 4.2% mannose was determined for the modified gelatin sample according with the area of the NMR signals. 

Further, a study with a water-soluble lectin (Con A) was also explored in order to evaluate the presence of mannose residues on the micellar surface after the incorporation of Gel(man) to the Soluplus^®^ micellar dispersion. Con A selectively binds to α-D-mannosyl and α-D-glucosyl residues due to its tetrameric form (with four independent sugar-binding sites) above pH = 7.0 [40]. In this way, the presence of mannose residues on the surface of the nanoformulation would lead to the formation of large aggregates in comparison with the micellar system without the addition of mannose. Indeed, this lectin has been explored to confirm the surface mannosylation of different nano-sized carriers [41].

As could be observed in Table 1, before the addition of Con A, Soluplus^®^ micelles and Soluplus^®^ (man)micelles (dispersed in phosphate buffer pH 7.2) demonstrated unimodal size distributions of 158.1 ± 3.5 nm and 270.1 ± 31.3 nm, respectively. Furthermore, the PDI values observed were 0.205 and 0.275 for the micellar systems with or without mannose, respectively.

However, after the incubation with the lectin, a change in the size distribution pattern for the Soluplus^®^ (man)micelles could be observed. In this case, a bimodal size distribution with a main size peak of 265.5 ± 5.1 nm and a second size population of 4370 ± 392 nm was observed, along with an increment in the polydispersity of the sample (PDI 0.359), suggesting the formation of large aggregates due to the presence of mannose moieties on the micellar surface.

This change of the size distribution was not observed for those Soluplus^®^ micelles without the addition of mannose. In this case, there was still only one size population (161.8 ± 4.6 nm) with a narrow size distribution (PDI 0.201) after Con A incubation. Similar results were observed after the incubation of Con A with mannosylated PEG-based polymeric micelles [41].

### 3.2. Micellar Preparation, Drug Encapsulation and Physicochemical Stability under Dilution

Polymeric micelles (PMs) are amphiphilic, self-assembled nanocarriers that represent an attractive platform for drug delivery [42]. Due to self-aggregation in water, which occurs above the critical micellar concentration (CMC), PMs are characterized by a hydrophobic core and a hydrophilic corona, which provide both stability to the nanocarrier (due to hydrophilic corona) and lipophilic drug encapsulation (within their hydrophobic core) [17,22,42]. One of the biomaterials highlighted for PMs preparation is known as Soluplus^®^. Previously, we successfully demonstrated that this biopolymer is able to encapsulate hydrophobic drugs as RIF [20], CUR and PTX [17,22] (conforming simple and mixed micelles) to optimize tuberculosis and cancer therapy. These micellar systems exhibited excellent colloidal properties combined with high physicochemical stability under dilution. The last beings of great clinical importance since micellar systems are dynamic colloidal nanocarriers, which could undergo dis-assembling upon dilution after administration [43]

In particular, a Soluplus^®^-based micellar (3% *w*/*v*) nanoformulation with RIF (10 mg/mL) was developed as a respirable nanocarrier with an optimized in vitro microbicidal efficacy versus a RIF solution [20]. In this framework, we aimed to expand the potential of this nanoformulation by the co-encapsulation with an antioxidant drug as CUR and the surface decoration of the nanocarrier with mannose residues for an active drug targeting macrophages. In this case, CUR was incorporated due to its potential anti-mycobacterial properties associated with the host immune response modulation [18].

In a first step, we successfully developed a Soluplus^®^ micellar system (10% *w*/*v*) within RIF (10 g/mL) and CUR (5 mg/mL) employing a solvent-diffusion technique. Samples remain translucent to the naked eye with a bright red color due to the presence of RIF and CUR without the presence of any aggregates (Figure 3). It is worth stressing that an increment of the Soluplus^®^ concentration from 3% *w*/*v* to 10% *w*/*v* was required for the co-encapsulation of both RIF and CUR. Then, the RIF concentration was 10 mg/mL since it is clinically relevant for a pulmonary dosage form taking into account that the RIF concentration in an inhalable formulation could be approximately 100-fold lower than in an oral dosage form [44].

Furthermore, it was observed that 5 mg/mL of CUR was the highest drug concentration that could be employed in order to maintain the colloidal stability of the micellar system. Higher CUR concentrations led to micellar aggregation and drug precipitation over time (data not shown).

The DLS analysis at 25 °C showed that this nanoformulation exhibited an adequate colloidal stability with one size population and a narrow size distribution before (82.2 ±1.7 nm; PDI: 0.221) and after (81.2 ± 2.1 nm; PDI: 0.138) freeze-drying. A similar trend was observed for the drug-free micellar systems (Table 2). It is worth stressing that the drug incorporation within the micellar core did not lead to an increase of the micellar size. This behavior may be connected to the dynamic nature of the PMs where the final hydrodynamic diameter is mainly associated with the combination of biomaterial and the encapsulated drugs. Similar results were observed after CUR encapsulation in paclitaxel-loaded Soluplus^®^/TPGS mixed micelles [17]. At this first approach, micellar nanoformulations showed good post lyophilization aspects, with easy redispersion in distilled water without any macroscopic aggregates.

In a second step, based in previous investigations employing nanocarrier coating with a mannose-based surfactant for a respirable nanoformulation [45], we performed a pre-formulation assay in order to determine the amount of Gel(man) (0.25–1% *w*/*v*) that could be employed to surface-coating the micellar system without affecting its colloidal stability in terms of size and size distribution.

For those systems in the absence of RIF and CUR, it could be observed that the coating with mannose led to an increment of the micellar size and PDI values. For instance, micellar size increased from 114.1 ± 1.2 nm (without mannose) to 287.6 ± 28.3 nm (Gel(man) 0.7% *w*/*v*) and 345.0 ± 68.7 nm (Gel(man) 1% *w*/*v*) before lyophilisation. A similar trend was followed by the PDI values (Table 2). It is worth stressing that the Dh determination includes the micellar corona along with its associated solvent molecules from the external medium. Hence, the addition of hydrophilic additives as polymers could interact with the micellar corona, according with their water affinity, and this interaction would influence the final Dh values. An example of this kind of interaction is the addition of glycols and PEG 400 to poloxamers and poloxamines [46].

Interestingly, after the freeze-drying process, a decrement of the polydisperity and the micellar size of the nanoformulations with PDI values between 0.207 and 0.257 was observed for those systems with a higher (between 0.7% and 1% *w*/*v*) content of Gel(man). This effect could be related to a micellar contraction phenomenon after the lyophilization process due to the dynamic nature of the PMs. With lower Gel(man) concentrations, the lyophilisation process did not alter either the PDI values or the micellar size (Table 2).

These results demonstrated that the colloidal stability of Soluplus^®^ (man)micelles could be enhanced after freeze-drying. In this way, higher amounts of Gel(man) could be incorporated to actively target the cellular reservoirs of the Mtb.

A similar trend was observed after Gel(man) addition to the co-loaded (RIF and CUR) micellar dispersions before lyophilisation. The increment of Gel(man) concentration led to an increment of both micellar size and size distribution with only one size population. For instance, the micellar size increased from 122.2 ± 3.7 nm to 360.2 ± 11.2 nm for 0.25% *w*/*v* and 1% *w*/*v* of Gel(man), respectively. Nevertheless, after freeze-drying, only those micellar dispersions with the higher Gel(man) content (1% *w*/*v*) showed a sharp increment of micellar size and polydispersity (size and PDI). This could be associated with micellar aggregation and drug precipitation over time.

On the contrary, Soluplus^®^ (man)micelles with Gel(man) concentrations between 0.25% and 0.7% *w*/*v* demonstrated lower hydrodynamic diameters and narrow PDI values than their counterparts before freeze-drying (Table 2).

Overall, the drug-co-loaded micellar nanoformulation with the higher Gel(man) concentration (0.7% *w*/*v*) that demonstrated an adequate colloidal stability after freeze-drying was chosen for further analysis.

Taking into account the dynamic nature of the PMs, we aimed to further predict their aggregation behavior after pulmonary administration. To reach this objective, we diluted (1/50) the drug-loaded nanoformulations with Simulated Interstitial Lung Fluid (SILF) at 37 °C.

Initially, as could be observed in Table 3, there was a decrement of the Dh values after the PMs dilution in SILF in comparison with the un-diluted systems in distilled water (Table 2). This reduction of the micellar size could be expended due to the dynamic equilibrium between copolymer monomers in the bulk and those being part of the polymeric micelles. Upon dilution, the number of monomers of the micelles could decrease in order to maintain the free amphiphilic molecules of the bulk. Then, a reduction of the micellar size could be observed [47].

On the other hand, both micellar systems (in the absence and presence of mannose) demonstrated colloidal stability after their dilution in SILF over time, as only one size population with a narrow size distribution was detected over 24 h. Moreover, there was only a slight size increase from 56.04 ± 0.48 nm (0 h) to 62.69 ± 0.45 nm (24 h) for RIF–CUR-micelles. A similar trend was followed by the RIF–CUR-(man)micelles where micellar size increased from 61.26 ± 0.65 nm (0 h) to 68.13 ± 1.10 nm (24 h). In this framework, these results showed that the micellar dispersions (regardless of the presence of mannose) exhibited colloidal stability under dilution, standing as a potential nanoformulation for pulmonary administration.

Finally, a morphological characterization of the RIF (10 mg/mL)-CUR (5 mg/mL)-(man) (0.7% *w*/*v*) micelles (10% *w*/*v*) was assessed by TEM visualization. As can be observed in Figure 3, micelles demonstrated a spherical shape as previously described for Soluplus^®^-based PMs [20]. This morphology provides to the nanoformulation a great potential for drug intracellular accumulation since spherical colloidal nanocarriers could be easily up-taken by macrophages [48].

### 3.3. CMC Determination

The CMC value is the concentration at which amphiphilic molecules occupy the total of the air/water interface and start forming micelles. In the present study, the CMC was detected from a graph in which the derived count rate as a function of Soluplus^®^ or Soluplus^®^(man) (0.7% *w*/*v*) concentration (% *w*/*v*) was plotted, where the CMC value (% *w*/*v*) was observed as having a sharp increase in the derived count rate.

No difference in CMC value was observed for either treatment. The CMC value for drug-free micelles was 0.000541% *w*/*v,* and the CMC value for drug-free (man)-micelles was 0.000495% *w*/*v*. In this case, low CMC values were observed, which ensures micellization even under diluted conditions [47].

### 3.4. Nebulization Studies

Micelles containing RIF, CUR or both, RIF–CUR-(man)micelles, along with the corresponding drug dispersions, were nebulized using an air jet nebulizer (PariSX^®^) connected to the NGI, and both in vitro deposition and aerodynamic diameter of droplets were measured to evaluate their suitability and effectiveness as pulmonary delivery systems. As can be seen in Table 4, the aerodynamic diameter of all nanosystems was ≤3 μm, without significant differences between the micellar systems and the drug dispersions, suggesting that during the nebulization process, the device was capable of forming small droplets with the appropriate size for pulmonary administration. Indeed, as previously reported, the devices mainly used for pulmonary administration may generate particles with different aerodynamic diameters, but only the smaller ones (MMAD between 1 and 5 μm) are deposited by gravitational settling in the deeper part of the lungs, while those with MMAD > 5 μm are generally recovered in the upper airways (mouth, trachea and main bronchi) by inertial impaction [27].

As previously reported [49], the process of jet nebulization involves repeated cycles of aerosol formation and recapture in the nebulizer reservoir before the sample leaves the device, during which shearing forces are applied to the tested formulations, and for this reason, the carrier characteristics and stability play a key role for its use as a system for pulmonary administration. The nebulizer content, by using both nanosystem and dispersions, was not completely aerosolized being that the highest (~100%) total mass output (TMO) was obtained by nebulizing RIF-micelles, while the lowest (~81%) was by nebulizing RIF–CUR-(man)micelles (Table 4). This is probably due to the higher viscosity of the last system, which reduces the efficiency of the device along with the amount of drug nebulized. Any significant differences were detected between nanosystems and dispersions, suggesting the good aptitude of both to be nebulized. However, the suitability of RIF, CUR or both to be nebulized was improved by their incorporation into the nanosystems, as only the formulations containing Soluplus^®^ alone or combined with Gel(man) were capable of reaching the latest stages of the impactor in a high amount and percentage, which mimic the deeper airways. Indeed, the FPD and the FPF obtained by nebulizing the dispersions was lower and always less than 50% (Table 4). Overall results suggest that the micellar formulations, especially RIF–CUR-(man)micelles, seem to be ideal systems capable of effectively delivering both drugs to the deep lung.

### 3.5. In Vitro Antioxidant Capacity

An experimental Mtb-infection model in guinea pigs demonstrated oxidative stress conditions associated to endothelial damage mediated by free radicals and an insufficient serum total antioxidant capacity, similarly to the human TB disease [50]. On this matter, CUR properties could contribute to adverse lung effects associated with TB infection together with RIF antibiotic activity. As expected, the antioxidant activity of Cur was very high since its capability of scavenging a variety of reactive oxygen species including superoxide anion radical, hydrogen peroxide, hydroxyl radical, singlet oxygen, nitric oxide and other organic free radicals is well known [17]. Indeed, the addition of an antioxidant therapy to the conventional TB treatment could improve the disease outcome [50]. Hence CUR co-encapsulation with RIF within the mannose-decorated micelles could represent a potential alternative to optimize pulmonary TB therapy.

Numerous methods are used to evaluate antioxidant activities of natural compounds in formulations, foods and biological systems. Two free radicals that are commonly used to assess antioxidant activity in vitro are 2,2-azinobis (3-ethylbenzothiazoline-6-sulfonic acid) (ABTS) and 2,2-diphenyl-1-picrylhydrazyl (DPPH). The ABTS assay measures the relative ability of antioxidants to scavenge the ABTS generated in the aqueous phase, while DPPH is a stable free radical used in organic phases [23].

In the present investigation, the total antioxidant capacity of the different formulations was assessed by monitoring their ability to scavenge ABTS˙ or DPPH. Every tested sample containing CUR was able to show antioxidant properties in both assays.

Interestingly, on ABTS˙ experiments, CUR dispersions (2239 ± 153 nmol Trolox Eq/mg, ABTS˙ inhibition 18.6 ± 1.6%) showed less antioxidant activity compared to the co-loaded micellar systems (23,075 ± 205 nmol Trolox Eq/mg, ABTS˙ inhibition ~100%) (Table 5). A similar behavior was observed for the RIF–CUR-(man)micelles (20,404.0 ± 663.0 nmol Trolox Eq/mg, ABTS˙ inhibition ~100%). For both micellar systems, with and without mannose, the total antioxidant capacity was significantly higher (*p* < 0.0001) than CUR and RIF dispersions (Table 5). The high water affinity of polymeric micelles on aqueous phases due to their hydrophilic corona allows CUR to exert its antioxidant properties, compared to poor solubility observed on CUR dispersions. Similarly, RIF solution showed antioxidant capacity (938 ± 212 nmol Trolox Eq/mg, ABTS˙ inhibition 5.5 ± 0.6%), which was significantly (*p* < 0.05) improved once incorporated into the PMs (3918 ± 104, ABTS˙ inhibition 16.0 ± 0.4%). Similar results were observed for the RIF + CUR dispersion (4523.0 ± 214 nmol Trolox Eq/mg, ABTS˙ inhibition 35.5 ± 1.7%) and the drug-co-loaded micelles in the absence and presence of mannose (Table 5).

In contrast, due to its high solubility on methanol phases, CUR was observed to have an increased antioxidant activity using the DPPH˙ test compared to ABTS˙ (2678 ± 193 nmol Trolox Eq/mg, DPPH inhibition 49.7 ± 7.4%). This antioxidant capacity was increased (3440.0 ± 5.7 nmol Trolox Eq/mg, DPPH inhibition 96.2 ± 0.7%) for the solution of RIF + CUR. Micelles containing CUR and RIF showed an increased activity compared to CUR solutions before (3633 ± 68 nmol Trolox Eq/mg, DPPH inhibition 75.6 ± 1.3%) and after (3302.0 ± 83.0 nmol Trolox Eq/mg, DPPH inhibition 61.6 ± 1.8%) mannose addition (Table 6). Once again, there was a significant increment in the total antioxidant activity of RIF (*p* < 0.0001) and CUR (*p* < 0.05) after their encapsulation within the polymeric nanocarriers versus free drugs (Table 6).

Furthermore, both assays demonstrated that the antioxidant properties of CUR were not affected by its co-encapsulation within PMs.

### 3.6. In Vitro Drug Release

To better understand the potential of the nanoformulations to deliver RIF and CUR to the lungs, the in vitro drug releases were evaluated in SILF at 37 °C over 24 h employing a dialysis method. 

On the one hand, results demonstrated that RIF could be successfully released from the polymeric matrix. For instance, 50.4% of RIF was release over 4 h along with a sustained and complete drug release (~100%) at 24 h for the RIF–CUR-micelles. Interestingly, the mannose surface decoration slightly slowed the RIF release from those micellar systems. As can be observed in Figure 4, there was a drug cumulative release of 46.08% after 4 h and 84.2% over 24 h. 

On the other hand, the release of CUR was significantly different from that of RIF, regardless of the presence of mannose. Indeed, a slowed drug release was observed in comparison with RIF over 24 h. For instance, after 8 h, 19.9% and 19.7 % of CUR was released in the release medium from Soluplus^®^ micelles and Soluplus^®^ (man)micelles, respectively. Then, after 24 h, CUR release was 56.2% and 51.1% from the micellar systems with and without mannose, respectively (Figure 4), confirming a reduced effect of mannose on the CUR release. These results suggested a high affinity between the lipophilic CUR and the hydrophobic micellar core, as previously observed after CUR encapsulation into PMs employing Soluplus^®^ [17].

### 3.7. Hemolysis Assay

For a respirable nanoformulation, it is important to investigate its hemolytic potential, since drugs will interact with the lung capillary network and the bloodstream after their pulmonary administration [51].

This is the main reason why the in vitro hemolytic performance of the RIF–CUR-(man)-micelles, its drug-free counterpart and a RIF solution were compared. The maximum drug concentration assayed (20 µg/mL) was higher than the therapeutic drug concentrations in blood [52].

As can be seen in Figure 5, there was no hemolytic effect with the micellar dispersions and the RIF solution at the different drug concentrations assayed. For instance, a hemolytic percentage of 0.61 ± 0.28 and 0.81 ± 0.28 was observed for RIF–CUR-(man)micelles and drug-free-(man)micelles, respectively, for the highest concentration of RIF (20 μg/mL) (Figure 5). Interestingly, the RIF solution exhibit a hemolytic percentage of 2.17 ± 0.09 at the same concentration (20 μg/mL) (Figure 5).

In this framework, every formulation (mannose-decorated micellar systems and drug solution) demonstrated to be safe (non-hemolytic), taking into account guidance for the in vitro hemolysis [53]. This guidance states that pharmaceutical formulations with hemolysis percentages below 10% are considered as non-hemolytic.

### 3.8. Micellar Lung Accumulation

Previously, we reported the lung accumulation over 24 h of Soluplus^®^ micelles (3% *w*/*v*) [20]. Hence, in order to investigate the fate of the (man)micelles (10% *w*/*v*) after their intratracheal administration in Wistar rats, micellar systems were radiolabeled with 99mTc. For this purpose, a non-invasive technique was employed by means of acquiring radioisotopic planar images. Before the in vivo biodistribution assays, it was confirmed that the micellar radiolabeling was efficient (98%) where the concentration of predictable impurities remains under accepted limits [54]. Furthermore, it was observed that radiochemical purity was >95% in serum samples, denoting no release of 99mTc over 24 h

Herein, biodistribution assays demonstrated that although the technical procedure was performed carefully to introduce the 99mTc-(man)micelles in the trachea, and animals remained in a 45° position during the biodistribution time, some of the administered dose was regurgitated, and the biodistribution profile showed the results of two administration pathways: oral and intratracheal. For the intratracheal-administered (leftover) fraction (55% of the administered dose), 10 % was still at the administered point with tracheal deposition, since no rinse was made following administration, and 45% reached the lungs after 1 h of the administration (Figure 6A). The other regurgitated fraction (45% of the administered dose) was distributed in the stomach (15%) and intestines (30%) by the same time (Figure 6A). Interestingly, after 24 h, the radiolabeled micellar system washed out from the trachea towards the lungs, where it remained by 53% of lung deposition and no tracheal accumulation (Figure 6B). On the other hand, stomach and intestines uptake drastically decreased over 24 h (up to 15%) due to fecal elimination. Altogether, the results showed that almost 100% of the 99mTc-(man)micelles that reached the lungs remained there 24 h after the administration.

In this framework, the potential as an inhalable nanoformulation for TB therapy is highlighted due to the ability of the micellar systems to accumulate in lungs over time. 

### 3.9. Microbicidal Efficacy against Mycobacterium tuberculosis

After developing a colloidal RIF–CUR-based micellar nanoformulation, which could be nebulized to reach the deep lung and accumulated there over 24 h with non-hemolytic effects, we aimed to evaluate its in vitro microbicidal efficacy. For this purpose, we employed derived macrophages (THP-1) infected with Mtb (H37Rv strain) by means of multiplicity of infection (MOI, 10). Samples were diluted to obtain a RIF final concentration of 5 µg/mL, since this is a comparable RIF concentration to therapeutic drug blood concentrations [52].

As it could be observed in Figure 7, the drug-free-micelles did not decrease the colony forming units (CFU) of infected macrophages, suggesting that the biomaterial did not exhibit microbicidal efficacy against Mtb H37Rv. Nevertheless, there was a decrement of CFU after macrophage incubation with RIF-micelles. These results are being in good concordance with previous studies employing RIF-loaded Soluplus^®^ micelles (3% *w*/*v*) [20]. Furthermore, the addition of CUR to the micellar system demonstrated a significant decrement (2.0-fold, *p* < 0.05) of the CFU versus RIF-micelles. These results suggest that the presence of CUR enhanced the antimicrobial efficacy of the nanoformulation, which is probably related to CUR’s capacity to induce caspase-3-dependat apoptosis and autophagy in derived THP-1 macrophages [55]. Similar results were assessed after co-loading of RIF and CUR in polyethylene sebacate nanoparticles for oral administration [19].

Furthermore, promising results were observed after the mannose coating of the RIF–CUR-micelles. In this case, a significant decrement (*p* < 0.05, 5.2-fold) of the CFU in comparison with their mannose-free counterpart was observed. Furthermore, an approximately 10-fold decrement of the CFU was observed for the mannosylated systems versus RIF-micelles (Figure 7). Similar results were observed for mannosylated nanoparticles covalently linked with isoniazid where the mannosylation strategy effectively promoted the complete bacterial eradication in comparison with the free drug and mannose-free nanoparticles [56]. Furthermore, mannosylated solid lipid nanoparticles loaded with isoniazid also demonstrated an enhanced intracellular antibiotic efficacy in vivo [57]. In this way, previous studies have demonstrated that mannosylation improves the macrophage uptake of solid lipid nanoparticles associated with the presence of the membrane mannose receptor in human THP1 macrophages [58].

The overall results highlight the potential of our mannosylated nanoformulation for an active drug targeting macrophages, leading to an improved inhalable anti-TB therapy. 

## 4. Conclusions

A respirable co-loaded micellar system for an active RIF and CUR delivery to macrophages was successfully developed. To the best of our knowledge, this is the first time that RIF and CUR were co-loaded in a mannose surface-decorated micellar system for pulmonary administration.

The micellar dispersion coated with mannose demonstrated an excellent colloidal stability even after dilution in SILF. Furthermore, the antioxidant properties of CUR were not affected by its encapsulation within PMs. Our nanoformulation did not exhibit hemolytic potential, and it was suitable for nebulization and drug delivery to the deep lung, according to its aerodynamic diameter. In vivo biodistribution studies confirmed the lung accumulation of the radiolabeled PMs over 24 h. Furthermore, the in vitro microbicidal efficacy against Mtb H37Rv was clearly enhanced after mannose coating. In this context, the potential of our mannose-coated RIF–CUR-nanoformulation for an optimized pulmonary anti-TB therapy was confirmed. Further studies employing an in vivo TB-infection model will be performed to investigate the anti-TB performance of this nanoformulation to optimize TB pulmonary therapy.

## Figures and Tables

**Figure 1 pharmaceutics-14-00959-f001:**
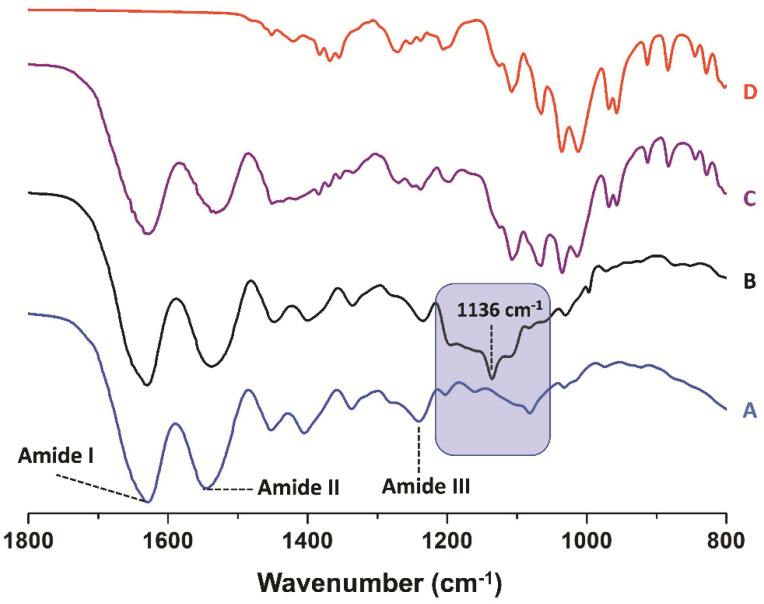
ATR-FTIR spectra for gelatin (A), mannose-modified gelatin (B), physical mixture of gelatin with 5% mannose (C) and mannose (D).

**Figure 2 pharmaceutics-14-00959-f002:**
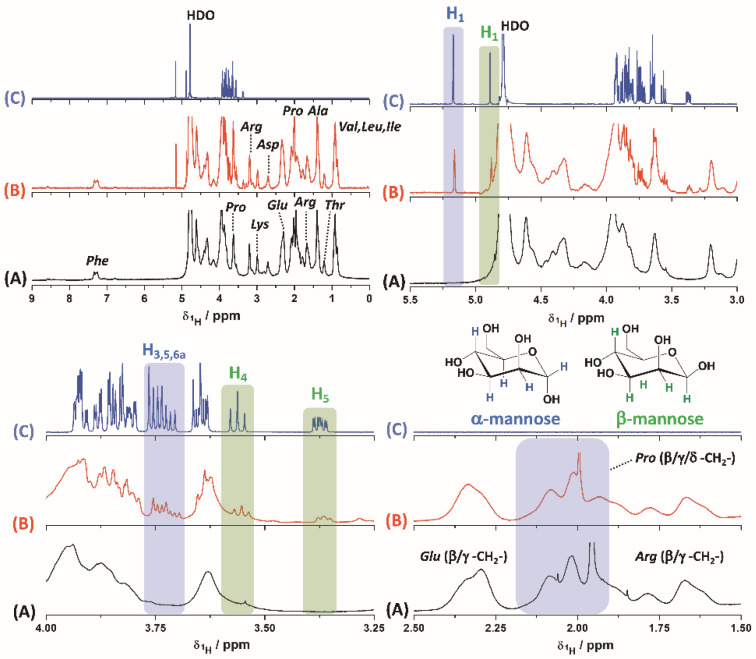
^1^H-NMR spectra for gelatin (A), mannose-modified gelatin (B) and mannose (C) dissolved in D_2_O. Different regions are shown for each of the samples for better understanding. The hydrogens of the α- and β-anomers for D-mannose are drawn with different colors for the assignments of the NMR signals.

**Figure 3 pharmaceutics-14-00959-f003:**
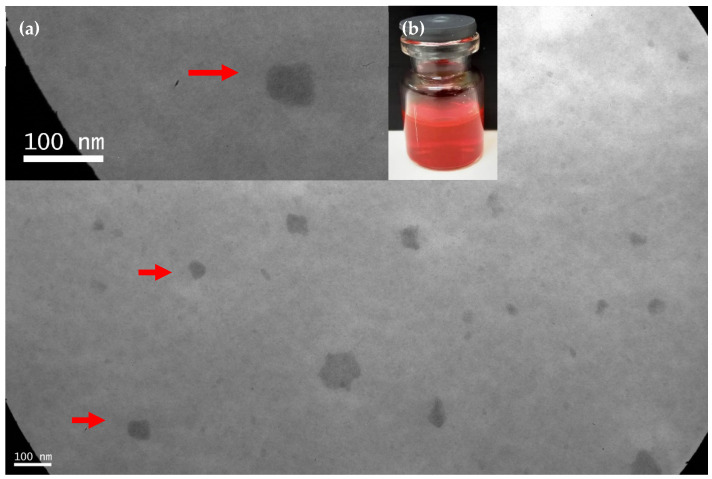
TEM micrograph of RIF–CUR (10 mg/mL and 5 mg/mL)-(man)(0.7% *w*/*v*)-micelles (10% *w*/*v*). Red arrows point out the polymeric micelles. Scale bar: 100 nm. Photo Inset: (**a**) Magnification of TEM micrograph and (**b**) macroscopic aspect of the drug-loaded micellar dispersion after re-dispersion in distilled water.

**Figure 4 pharmaceutics-14-00959-f004:**
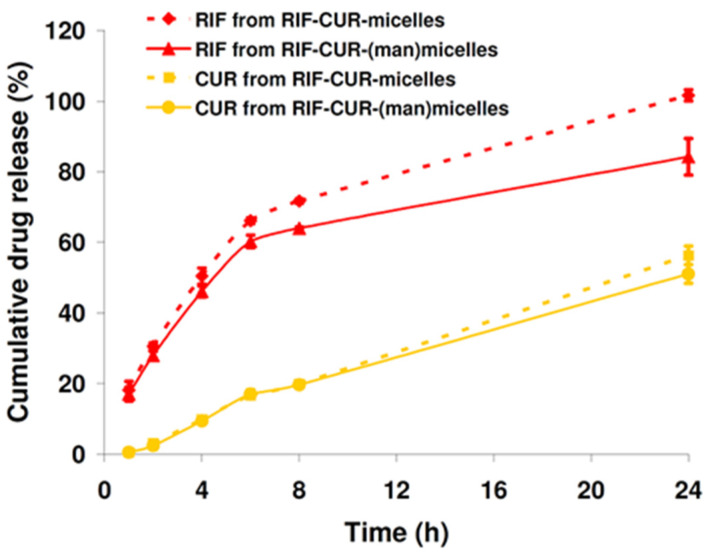
In vitro RIF and CUR release profiles from RIF–CUR-micelles and RIF–CUR-(man)micelles (10% *w*/*v*) at pH 7.4 (SILF, 37 °C) over 24 h. Results are expressed as mean ± standard deviation (S.D.) (n = 3). RIF and CUR concentrations were 10 mg/mL and 5 mg/mL, respectively.

**Figure 5 pharmaceutics-14-00959-f005:**
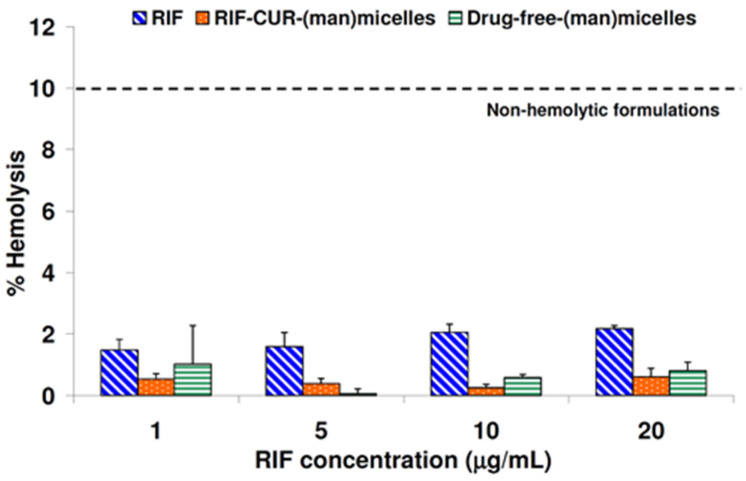
Percentages of hemolysis of RIF, drug-free-(man)micelles and RIF–CUR-(man)micelles at various concentrations after incubation for 3 h at 37 °C. Results are expressed as mean ± S.D. (n = 3).

**Figure 6 pharmaceutics-14-00959-f006:**
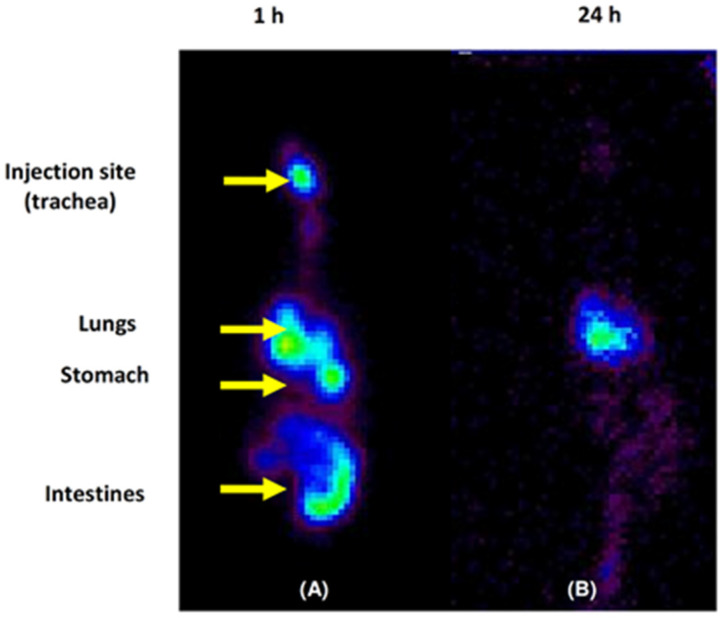
Biodistribution of 99mTc radiolabeled Soluplus^®^ (man)micelles (1.85 MBq). Static images were acquired 1 (**A**) and 24 h (**B**) post intratracheal administration by means of a surgical puncture tracheotomy. Anesthesia: isofluorane. After 24 h post-administration, almost 100% of the 99mTc (man)micelles of the intratracheal route remained in the lungs (color intensity in the image is not corrected by physical decay of the signal).

**Figure 7 pharmaceutics-14-00959-f007:**
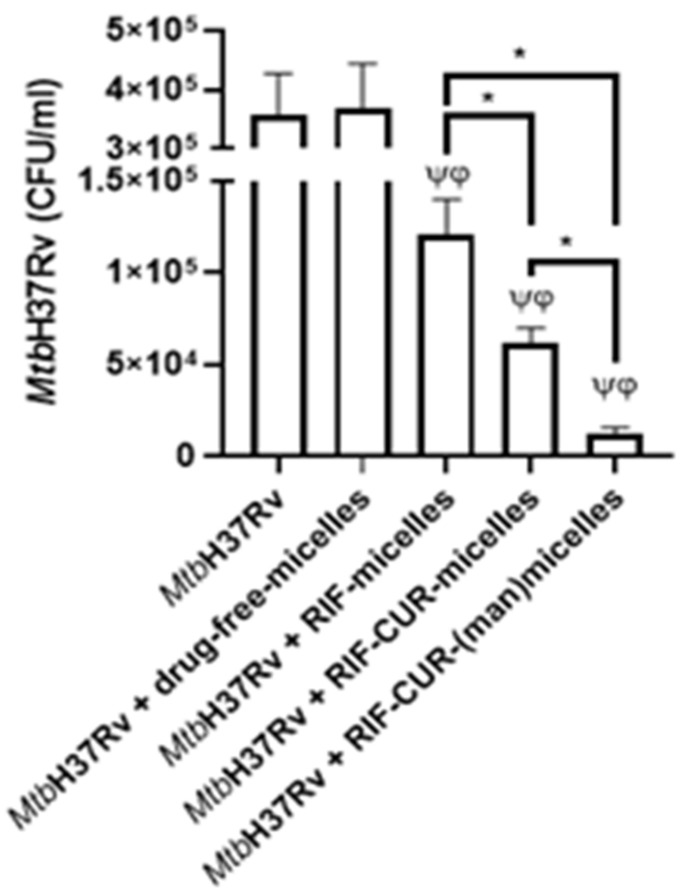
Intracellular survival of *Mycobacterium tuberculosis* H37Rv in RIF-(10 mg/mL) micelles (10% *w*/*v*), RIF–CUR (10 mg/mL and 5 mg/mL)-micelles, RIF–CUR (10 mg/mL and 5 mg/mL)-(man)micelles and drug-free-micelles treated THP-1 cells. Macrophages derived from THP-1 cells line (1 × 106 cells/mL) were infected with Mtb H37Rv (MOI: 10). After 2 h of infection, the culture medium was replaced, and cells were cultured with RIF-micelles (5 μg/mL), RIF–CUR-micelles (5 μg/mL and 2.5 μg/mL) or RIF–CUR-(man)micelles (5 μg/mL and 2.5 μg/mL) for 48 h. Then, cells were washed and lysed for mycobacterial colony-forming units (CFU) determination. Data are presented as means of bacterial viability (CFU expressed as percentage of the control) ± standard error of the mean (SEM), * *p* < 0.05; φ *p* < 0.001; ψ *p* < 0.001. *p* values were calculated using one-way ANOVA with post hoc Tukey’s multiple comparisons test.

**Table 1 pharmaceutics-14-00959-t001:** Micellar size and size distribution (PDI) of Soluplus^®^ micelles and Soluplus^®^ (man)micelles (10% *w*/*v*) in presence and absence of Con A after incubation (2 h) at 25 °C. Data are expressed as mean ± S.D. (n = 5).

			Dh (nm) (±SD)					
Sample	Con A Presence	PDI (±SD)	Peak 1	Intensity (%)	Peak 2	Intensity (%)	Peak 3	Intensity (%)
Soluplus^®^ micelles	-	0.205 (0.006)	158.1 (3.5)	100.0	-	-	-	-
	+	0.201 (0.005)	161.8 (4.6)	100.0				
Soluplus^®^ (man)micelles	-	0.275 (0.013)	270.1 (31.3)	100.0	-	-	-	-
	+	0.359 (0.008)	246.5 (5.1)	94.2	4370 (392)	5.3	11.2 (19.4)	0.6

**Table 2 pharmaceutics-14-00959-t002:** Micellar size and size distribution (PDI) of free and drug-loaded Solupus micelles (10% *w*/*v*) with and without mannose at 25 °C, before and after lyophilization. Data are expressed as mean ± S.D. (n = 5).

Nanoformulation	Gel(man) Concentration (%)	Before Lyophilization		After Lyophilization	
		Size (nm) (±S.D.)	PDI (±S.D.)	Size (nm) (±S.D.)	PDI (±S.D.)
Drug-free-micelles	-	114.1 (1.2)	0.194 (0.002)	124.0 (1.3)	0.217 (0.006)
Drug-free-(man)micelles	0.25	137.2 (4.1)	0.213 (0.009)	127.4 (0.3)	0.257 (0.006)
	0.5	149.1 (1.8)	0.226 (0.003)	145.6 (4.4)	0.264 (0.007)
	0.7	287.6 (28.3)	0.414 (0.015)	164.5 (6.7)	0.257 (0.009)
	1.0	345.0 (68.7)	0.444 (0.003)	120.6 (4.7)	0.207 (0.009)
RIF–CUR-micelles	-	82.2 (1.7)	0.221 (0.007)	81.2 (2.1)	0.138 (0.020)
RIF–CUR-(man)micelles	0.25	122.2 (3.7)	0.224 (0.003)	113.7 (1.7)	0.215 (0.011)
	0.5	147.8 (4.6)	0.244 (0.010)	101.8 (3.8)	0.218 (0.009)
	0.7	197.6 (7.4)	0.312 (0.042)	108.1 (0.9)	0.208 (0.001)
	1.0	360.2 (11.2)	0.288 (0.004)	366.7 (5.5)	0.424 (0.011)

**Table 3 pharmaceutics-14-00959-t003:** Micellar size and size distribution of the RIF(10 mg/mL)-CUR (5 mg/mL)-loaded micelles (10% *w*/*v*) with and without mannose at 37°C in simulated interstitial lung medium (SILF, pH 7.4, sample dilution 1/50) over 24 h. Data are expressed as mean ± S.D. (n = 5).

Nanoformulation	Time (h)	Size (nm) (±S.D.)		PDI (±S.D.)
		Peak 1	Intensity (%)	
RIF–CUR-micelles	0	56.04 (0.48)	100.0	0.044 (0.007)
	1	58.15 (0.51)	100.0	0.040 (0.008)
	2	58.09 (0.08)	100.0	0.042 (0.009)
	3	57.49 (0.51)	100.0	0.037 (0.011)
	24	62.69 (0.45)	100.0	0.056 (0.008)
RIF–CUR-(man)micelles	0	61.26 (0.65)	100.0	0.103 (0.016)
	1	61.35 (0.27)	100.0	0.068 (0.008)
	2	61.44 (0.36)	100.0	0.075 (0.011)
	3	60.08 (0.37)	100.0	0.063 (0.004)
	24	68.13 (1.10)	100.0	0.080 (0.009)

**Table 4 pharmaceutics-14-00959-t004:** Total mass output (TMO%), fine particle dose (FPD, mg), fine particle fraction (FPF, %) and mass median aerodynamic diameter (MMAD) of nanosystems nebulized by using the next generation impactor (NGI). FPD and FPF values are shown as mean ± S.D. of three experiments; MMAD values are shown as mean ± geometric standard deviation.

Sample	Total Mass Output (%)	Fine Particle Dose (FPD) (mg)	Fine Particle Fraction (FPF) (%)	Aerodynamic Diameter (± Geometric Standard Deviation)
RIF	86 ± 6	8 ± 3	46 ± 4	1.27 ± 1.18
CUR	88 ± 7	2 ± 0.5	19 ± 3	2.18 ± 1.70
RIF + CUR	89 ± 12	8 ± 2	30 ± 4	2.52 ± 1.88
RIF-micelles	100 ± 3	15 ± 2	75 ± 15	1.26 ± 1.17
CUR-micelles	89 ± 2	5 ± 1	57 ± 6	2.14 ± 1.68
RIF–CUR-micelles	83 ± 6	15 ± 3	62 ± 9	1.65 ± 1.41
RIF–CUR-(man)micelles	81 ± 7	13 ± 2	52 ± 6	2.10 ± 1.66

**Table 5 pharmaceutics-14-00959-t005:** Antioxidant activity of nanoformulations (drug-free, RIF-micelles, RIF–CUR-micelles and RIF–CUR-(man)micelles) and drugs dispersion (CUR, RIF and RIF + CUR), calculated as the inhibition percentage of ABTS radical. Data are expressed as mean ± S.D. (n = 6).

Sample	Total Antioxidant Capacity (nmol Trolox Eq/mg Sample)	ABTS˙ Inhibition (%)
RIF	938.0 ± 212.0	5.5 ± 0.6
CUR	2239.0 ± 153.0	18.6 ± 1.6
RIF+ CUR	4523.0 ± 214.0	35.5 ± 1.7
Drug-free micelles	ND	ND
RIF-micelles	3918.0 ± 104.0 #	16.0 ± 0.4
RIF–CUR-micelles	23,075.0 ± 205.0 *	99.0 ± 1.0
RIF–CUR-(man)micelles	20,404.0 ± 663.0 *	99.0 ± 6.0

Note: Multiple comparisons were performed using one-way ANOVA and Tukey’s multiple comparisons post-hoc test. ND: not detected. * *p* < 0.0001 vs. RIF, CUR, drug-free micelles and RIF-micelles. # *p* < 0.05 vs. RIF.

**Table 6 pharmaceutics-14-00959-t006:** Antioxidant activity of nanoformulations (drug-free, RIF-micelles, RIF–CUR-micelles and RIF–CUR-(man)micelles) and drugs solutions (CUR, RIF and RIF + CUR), calculated as the inhibition percentage of DDPH radical. Data are expressed as mean ± S.D. (n = 6).

Sample	Total Antioxidant Capacity (nmol Trolox Eq/mg Sample)	DPPH˙ Inhibition (%)
RIF	595.0 ± 76.0	18.6 ± 2.9
CUR	2678.0 ± 193.0 #	49.7 ± 7.4
RIF + CUR	3440.0 ± 5.7	96.2 ± 0.7
Drug-free micelles	ND	1.0 ± 0.4
RIF-micelles	120.0 ± 37.0	25.2 ± 5.6
RIF–CUR-micelles	3633.0 ± 68.0 *	75.6 ± 1.3
RIF–CUR-(man)micelles	3302.0 ± 83.0 *	61.6 ± 1.8

Note: Multiple comparisons were performed using one-way ANOVA and Tukey’s multiple comparisons post-hoc test. ND: not detected. * *p* < 0.0001 vs. RIF, drug-free micelles and RIF-micelles. # *p* < 0.05 vs. RIF–CUR-micelles and RIF–CUR-(man)micelles.

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
