# Peer review of "Inhalable Mannosylated Rifampicin–Curcumin Co-Loaded Nanomicelles with Enhanced In Vitro Antimicrobial Efficacy for an Optimized Pulmonary Tuberculosis Therapy"

_pharmaceutics, 2022, doi:10.3390/pharmaceutics14050959_

Round 1

Reviewer 1 Report

In this manuscript, Galdoporpora and colleagues investigated the development of mannosylated micelles loaded with both rifampicin and curcumin for the treatment of pulmonary tuberculosis. This study indicates that these micelles were suitable for drug delivery to the deep lung and did not show haemolytic potential. The conjugation of mannose to these micelles improved by 5.2-fold the microbicidal efficacy against Mtb H37Rv of the drug-co-loaded systems compared with the non-targeted micelles.

The manuscript was very interesting to read. Some points will have to be clarified, as described below:

Abstract

  1. Could you please remove all abbreviations from the abstract
  2. Line 24: could you please correct “nanotechnological”
  3. Line 27: could you please correct “Mannose-coated”
  4. Line 29: could you please add the SEM to the micelle size mean
  5. Line 31: could you please correct “according with”

Introduction

  1. Line 41: “has been in the spot”: could you please reformulate this in a mores scientific way
  2. Line 47: “side-systemic effects”: could you please correct this
  3. Line 48: target (active targeting) the drug in a precise 48 site”: could you please reformulate this
  4. Line 49: “sugar-residues”: could you please replace this with “sugar residues”
  5. Line 92: “medicine10”: could you please correct the reference”
  6. Line 96: “nanatechnological”: could you please correct this”
  7. Line 104: “soluplus micelles”: could you please correct Soluplus® in the whole manuscript
  8. Line 101: could you please format “in vitro”, “in vivo”, the bacterial names and all latine names, in italic in the whole manuscript

Material and methods

  1. Lines 124-126: could you please provide more experimental details (amount of gelatin and mannose, amount of Soluplus and volume of phosphate buffer
  2. Line 129: “(3500 Da, Spectra/Por®3 Dialysis Membrane, molecular weight cut off = 3,500”: could you please indicate the MWCO only once
  3. Line 135: “ATR-FTIR 135 (diamond attenuated total reflectance) spectra obtained using a Nicolet iS50 Advanced 136 Spectrometer (Thermo Scientific).”: could you please reformulate this
  4. Line 141: “Si(CH3)4“: could you please correct this
  5. Line 159: “1×10−6”: could you please correct this
  6. Line 160: “before used”: could you please correct this
  7. Line 171: “RIF (100 mg) and CUR (50 mg) were 171 dissolved in acetone (20 mL): could you please provide additional data demonstrating that this was the optimal ratio of RIF and CUR to be loaded to the micelles. Could you please explain in the manuscript why you did not use 100 mg of RIF and 100 mg of CUR.
  8. Line 170: could you please describe how the micelles have been purified from the non-encapsulated drugs?
  9. Line 227: could you please move the statistical analysis at the end of the Material and Methods section
  10. Line 255: “MgCl2, 603.2 mg NaCl, 40.4 mg KCl, 43.6 255 mg Na2HPO4, 259.6 mg NaHCO3, 8.3 mg Na2SO4, 35.9 mg CaCl2, 60.7 mg CH3COONa”: could you please correct the formatting of the chemical formulas
  11. Line 292: “samples were centrifuged for 10 min at 3500 RPM”: could you please also indicate the speed of centrifugation in g
  12. Lines 371-373: could you please indicate the volume of PBS used for the washing step and the amount of oleic acid-albumin-dextrose-catalase

Results and discussion

  1. Lines 404, 417: “1H-NMR, “D2O”: could you please correct this
  2. Figure 1B: could you please further describe the results shown in Figure 1B
  3. Line 420: could you please correct the symbols used, in the whole manuscript
  4. Table 4: could you please correct “aerodynamic”
  5. Could you please describe the quantification of the drugs in the micelles
  6. Line 794: “Taking advance”: could you please reformulate this
  7. Line 799: could you please further discuss in the manuscript why the mannose presence promotes an active drug targeting to the 799 derived THP-1 macrophages

Author Response

In this manuscript, Galdoporpora and colleagues investigated the development of mannosylated micelles loaded with both rifampicin and curcumin for the treatment of pulmonary tuberculosis. This study indicates that these micelles were suitable for drug delivery to the deep lung and did not show haemolytic potential. The conjugation of mannose to these micelles improved by 5.2-fold the microbicidal efficacy against Mtb H37Rv of the drug-co-loaded systems compared with the non-targeted micelles.

The manuscript was very interesting to read. Some points will have to be clarified, as described below:

Abstract

  1. Could you please remove all abbreviations from the abstract

We thank the Reviewer for the suggestion, all the abbreviation in the Abstract have been deleted, accordingly.

  1. Line 24: could you please correct “nanotechnological”

We apologize for the mistake. Nanotechnological has been removed, accordingly.

  1. Line 27: could you please correct “Mannose-coated”

We apologize for the mistake. The term has been corrected accordingly

  1. Line 29: could you please add the SEM to the micelle size mean

Thanks, done accordingly.

  1. Line 31: could you please correct “according with”

We thank the Reviewer for the suggestion, “according with” has been corrected, accordingly

Introduction

  1. Line 41: “has been in the spot”: could you please reformulate this in a mores scientific way

We thanks the Reviewer for the suggestion, the sentence has been modified accordingly.

  1. Line 47: “side-systemic effects”: could you please correct this

We thank the Reviewer for the remark, the sentence has been corrected, accordingly.

  1. Line 48: target (active targeting) the drug in a precise 48 site”: could you please reformulate this

We thank the Reviewer for the suggestion, the sentence has been modified accordingly.

  1. Line 49: “sugar-residues”: could you please replace this with “sugar residues”

We thank the reviewer, sugar-residues has been replaced with sugar residues, accordingly

  1. Line 92: “medicine10”: could you please correct the reference”

Sorry for the mistake, the reference has been deleted (it was wrongly inserted) according to the Reviewer’s suggestion

  1. Line 96: “nanatechnological”: could you please correct this”

Sorry for the mistake, it has been corrected accordingly.

  1. Line 104: “soluplus micelles”: could you please correct Soluplus® in the whole manuscript

We thank for the remark, Soluplus has been corrected throughout the manuscript.

  1. Line 101: could you please format “in vitro”, “in vivo”, the bacterial names and all latine names, in italic in the whole manuscript

We thank the Reviewer for the remark, all the terms mentioned by the Reviewer have been formatted accordingly

Material and methods

  1. Lines 124-126: could you please provide more experimental details (amount of gelatin and mannose, amount of Soluplus and volume of phosphate buffer

We thank the Reviewer for the remark, we have detailed the method as requested

  1. Line 129: “(3500 Da, Spectra/Por®3 Dialysis Membrane, molecular weight cut off = 3,500”: could you please indicate the MWCO only once

We apologize for the mistake. As requested we have indicated the MWCO only once.

  1. Line 135: “ATR-FTIR 135 (diamond attenuated total reflectance) spectra obtained using a Nicolet iS50 Advanced 136 Spectrometer (Thermo Scientific).”: could you please reformulate this

We thank the Reviewer for the remark. As requested the sentence has been re-written.

  1. Line 141: “Si(CH3)4“: could you please correct this

Corrected accordingly.

  1. Line 159: “1×10−6”: could you please correct this

Sorry for the mistake, it has been corrected accordingly

  1. Line 160: “before used”: could you please correct this

The sentence has been corrected according to the Reviewer’ suggestion

  1. Line 171: “RIF (100 mg) and CUR (50 mg) were 171 dissolved in acetone (20 mL): could you please provide additional data demonstrating that this was the optimal ratio of RIF and CUR to be loaded to the micelles. Could you please explain in the manuscript why you did not use 100 mg of RIF and 100 mg of CUR.

The reviewer raises an interesting point. As described in the manuscript, we have previously developed a respirable micellar system based on Soluplus (3 % w/v) and rifampicin (10 mg/mL). This nanoformulation exhibited excellent colloidal stability along with a clinically relevant rifampicin concentration, especially because the RIF concentration in a respirable formulation could be approximately 100-fold lower than in an oral dosage form [S.K. Katiyar, S. Bihari, S. Prakash, Low-dose inhaled versus standard dose oral form of anti-tubercular drugs: concentrations in bronchial epithelial lining fluid, alveolar macrophage and serum, J. Postgrad. Med. 54 (2008) 245–246]. Hence, only 0.6 mL of this colloidal system would be required per day (rifampicin daily oral dose: 600 mg). This volume could be easily aerosolized taking into account a long-term anti-TB therapy [Grotz E. et al. Journal of Drug Delivery Science and Technology 53 (2019) 101170]. With this in mind, we aimed to further expand the potential of this colloidal nanocarrier by the addition of curcumin and the mannose coating.

As we developed this co-loaded micellar system, we aimed to maintain a clinically relevant rifampicin concentration of 10 mg/mL. Then, the amount of Soluplus® should be increased from 3% w/v to 10% w/v in order to co-encapsulate both drugs, rifampicin and curcumin. Further, we determined that the highest amount of CUR in the nanoformulation could be 5 mg/mL since higher amounts of curcumin led to nanocarrier aggregation (with higher PDI values) and precipitation over time.

As requested, we have discussed the reason why we employed 10 mg/mL of rifampicin and 5 mg/mL of curcumin in Section 3.2. Micellar preparation, drug encapsulation and physicochemical stability under dilution.

  1. Line 170: could you please describe how the micelles have been purified from the non-encapsulated drugs?

We understand the reviewer concern. On the one hand, rifampicin exhibits an amphiphilic nature and pH-dependant solubility in aqueous media. In the present investigation, the pH value of the co-loaded nanoformulations coated with mannose was 4.90 (Data not shown). Particularly, at low pH values (HCl 0.1N), it exhibits high aqueous solubility (200 mg/mL, 37°C) but it rapidly degrades into a non-active derivative known as 3-formyl-rifampicin. On the other hand, at pH values between 4.3 and 7.3, rifampicin exhibits lower aqueous solubility (1.3-2.5 mg/mL, 25°C) [G.G. Gallo, P. Radaelli, Analytical Profiles of Drug Substances, fifth ed., K. Florey, 1987, Moretton et al. Col. Surf. B: Biointerfaces 79 (2010) 467-479, Moretton et al. Nanomedicine (Lond) 9 (2014) 1635-1650].

Regarding curcumin, it also exhibits (bio)pharmaceutical limitations as its low-aqueous solubility (about 0.6 μg/mL) in pure water [Wu et al. Journal of Pharmaceutical and Biomedical Analysis 211 (2022) 114613] and poor bioavailability [Goel A.et al. Biochem. Pharmacol. 75 (2008) 787-809].

In this framework, a solvent-diffusion technique was employed in order to encapsulate both hydrophobic drugs within the polymeric micelles. This technique is one of the most investigated methods to prepare aqueous dispersions of polymeric micelles [Gaucher et al. J. Control. Release 109 (2005) 169-188; Owen et al. Nano Today (2012) 7, 53-65]. Hence, after acetone evaporation, the aqueous micellar dispersions were filtered (0.45 μm, acetate cellulose filters) in order to remove insoluble rifampicin and curcumin. Then, the concentrations of rifampicin and curcumin in the samples were determined by HPLC analysis.

To make this issue more clear, we have added some information to the following sentence in Section 2.4. Micellar preparation and drug encapsulation. “….and samples were filtered (0.45 μm, acetate cellulose filters, Microclar, Argentina) in order to remove the undissolved drugs”   

  1. Line 227: could you please move the statistical analysis at the end of the Material and Methods section

We thank the Reviewer for the suggestion; the statistical analysis section has been moved at the end of the Material and Methods section as recommended.

  1. Line 255: “MgCl2, 603.2 mg NaCl, 40.4 mg KCl, 43.6 255 mg Na2HPO4, 259.6 mg NaHCO3, 8.3 mg Na2SO4, 35.9 mg CaCl2, 60.7 mg CH3COONa”: could you please correct the formatting of the chemical formulas

We apologized for the mistake, the format of the chemical formulas has been corrected, as requested.

  1. Line 292: “samples were centrifuged for 10 min at 3500 RPM”: could you please also indicate the speed of centrifugation in g.

We thank the Reviewer for the suggestion; we have added the speed of centrifugation process as relative centrifugal force.

  1. Lines 371-373: could you please indicate the volume of PBS used for the washing step and the amount of oleic acid-albumin-dextrose-catalase

We apologized for the omission. We have added all the information requested.

Results and discussion

  1. Lines 404, 417: “1H-NMR, “D2O”: could you please correct this

We apologized for the mistake. It has been corrected as requested.

  1. Figure 1B: could you please further describe the results shown in Figure 1B

We thank the Reviewer for the suggestion. We have described in more detail Figure 1B.

  1. Line 420: could you please correct the symbols used, in the whole manuscript

We apologized for the mistake. It has been corrected as requested.

  1. Table 4: could you please correct “aerodynamic”

We apologized for the mistake. It has been corrected as requested.

  1. Could you please describe the quantification of the drugs in the micelles

We understand the reviewer point of view. In the present investigation, we have encapsulated within Soluplus polymeric micelles two hydrophobic drugs (rifampicin and curcumin) with poor-aqueous solubility. For micellar dispersions it has been stated that the amount of drug in the micelles can be quantified spectrophotometrically or via HPLC analysis after separation of the undissolved drug,  no differentiation is required between the encapsulated and the free drug due to the extremely low water solubility of hydrophobic drugs. However, a separation method (for instance using ultrafiltration spin columns) is recommended to determine the DL (%) and EE (%) of hydrophilic drugs encapsulation within polymeric micelles [Ghezzi M et al. Journal of Controlled Release 332 (2021) 312–336]. This is the main reason why we determined the drug concentration in the nanoformulation by RP-HPLC with UV detection (detection wavelength was 425 nm and 333 nm for curcumin and rifampicin, respectively).

  1. Line 794: “Taking advance”: could you please reformulate this

As requested we have reformulated the sentence as follows: “Furthermore, promising results were observed after the mannose coating of the RIF-CUR-micelles.”

  1. Line 799: could you please further discuss in the manuscript why the mannose presence promotes an active drug targeting to the 799 derived THP-1 macrophages

We understand the reviewer point of view. For a better understanding, we have re-writen the paragraph with a discussion of the in vitro efficacy results based on previous investigations employing the mannosylation strategy.

Reviewer 2 Report

Overall, the work is interesting, but I have some concerns.

1) I understand from the protocols that the interaction between mannose and gelatine is rather electrostatic than covalent. If so, the authors should demonstrate that such interaction is not going to weaken over time, since that could lead to non-targeted micelles. 

2) My main concern is that authors highlight the improved stability of the nanoformulation after freeze-drying. However, if they highlight that I assume the rest of the experiments should have been done following that protocol, because that would be how a potential marketed product from this nanomaterial would be administered. I think that at least the release experiment and the in vivo nebulization should be repeated. 

3) A control group is missing in the in vivo experiment. 

Author Response

1) I understand from the protocols that the interaction between mannose and gelatine is rather electrostatic than covalent. If so, the authors should demonstrate that such interaction is not going to weaken over time, since that could lead to non-targeted micelles. 

We understand the reviewer concern. As reported in the manuscript, the ATR-FTIR and 1H-NMR analysis demonstrated a molecular interaction between mannose and gelatine. However, the micellar dispersions were diluted from 10 mg/mL to 5 μg/mL to get a comparable RIF concentration to therapeutic drug blood concentrations in the in vitro microbicidal assay. This represents a sample dilution of approximately 1/2000. Then, results demonstrated that mannose coated micelles significantly (p<0.05) reduced the CFU of Mtb in comparison with the un-coated micelles. Then, the dilution of this mannose coated micelles did not affect their microbicidal effect. 

On the other hand, this nanoformulation was developed for a pulmonary administration where it has been stated that RIF concentration in a respirable formulation could be approximately 100-fold lower than that employed in an oral dosage form [S.K. Katiyar, S. Bihari, S. Prakash, Low-dose inhaled versus standard dose oral form of anti-tubercular drugs: concentrations in bronchial epithelial lining fluid, alveolar macrophage and serum, J. Postgrad. Med. 54 (2008) 245–246].The RIF dose per weight could be aerosolized employing a wide variety of nebulizers [Patil JS. et al. Lung India. 2012 Jan-Mar; 29(1): 44–49]. Indeed, the aerodynamic diameter of the nanoformulations was evaluated employing a PariSX® air jet nebulizer.

2) My main concern is that authors highlight the improved stability of the nanoformulation after freeze-drying. However, if they highlight that I assume the rest of the experiments should have been done following that protocol, because that would be how a potential marketed product from this nanomaterial would be administered. I think that at least the release experiment and the in vivo nebulization should be repeated. 

We thank the Reviewer for the remark. As reported in the manuscript, the freeze-drying effectively improves the stability of the dispersions. Given that, all the experiment (release and nebulization) have been performed starting from the powder (lyophilized), which was redispersed in water just before its use according to our idea of a stable product easily re-dispersible just before its administration by means of aerosol therapy.

3) A control group is missing in the in vivo experiment. 

We understand the reviewer point of view. We have previously observed that Soluplus micelles (3% w/v) exhibited lung accumulation over 24 h after intratracheal administration [Grotz E. et al. Journal of Drug Delivery Science and Technology 53 (2019) 101170]. Hence in the present study, we aimed to investigate the biodistribution of the Soluplus micelles after mannose coating to asses their potential for pulmonary administration. Moreover, this comparison with previous investigation allows reducing the number of animals for ethical reasons.

To clarify this issue, we have added the following sentence in Section 3.8 Micellar Lung Accumulation “Previously, we reported the lung accumulation over 24 h of Soluplus® micelles (3% w/v) [Grotz E. et al. Journal of Drug Delivery Science and Technology 53 (2019) 101170]”

Reviewer 3 Report

In their manuscript entitled "Inhalable mannosylated rifampicin-curcumin co-loaded nanomicelles with enhanced in vitro antimicrobial efficacy for an
optimized pulmonary tuberculosis therapy" the authors describe a very interesting research they performed.

One suggestion: Figure 3: the scale bar is not so easily to identify; As a suggestions, the Authors may also add a second TEM image (manification of the part with the "red arrows")  and insert in the figure to see better the scale bar and compare it to the particle.

This manuscript also fits perfectly to  the "world tuberculosis day" which was on 24. March 2022.

The manuscript is of an exeptional high quality with really high load of intensive work and experiments and interdisciplinarity-  and represents a very well planned research and analysis.

I can recommend it for publication.

Author Response

In their manuscript entitled "Inhalable mannosylated rifampicin-curcumin co-loaded nanomicelles with enhanced in vitro antimicrobial efficacy for an
optimized pulmonary tuberculosis therapy" the authors describe a very interesting research they performed.

One suggestion: Figure 3: the scale bar is not so easily to identify; As a suggestions, the Authors may also add a second TEM image (manification of the part with the "red arrows") and insert in the figure to see better the scale bar and compare it to the particle.

We thank the Reviewer for the suggestion. We have improved the Figure 3 as requested.

This manuscript also fits perfectly to  the "world tuberculosis day" which was on 24. March 2022.

The manuscript is of an exeptional high quality with really high load of intensive work and experiments and interdisciplinarity-  and represents a very well planned research and analysis.

I can recommend it for publication.

We thank the Reviewer for his positive comments.

Reviewer 4 Report

Comments: 

The authors have tried to synthesize inhalable mannosylated rifampicin-curcumin co-loaded nanomicelles with enhanced in vitro antimicrobial efficacy for an optimized pulmonary tuberculosis therapy and characterized in terms of micellar size, size distribution, and morphology. Moreover, the in vitro nebulization aptitude of formulations and their aerodynamic diameter were evaluated along with their hemolytic potential. Finally, the microbicidal effectiveness of formulations and the micellar accumulation in the lungs has been investigated in vitro by using THP-1 infected (Mtb H37Rv) macrophages and in vivo by using Wistar rats, respectively. The paper's subject is interesting, and the authors have tried performing several in vitro and in vivo experiments to support their experimental findings. While this study falls within the scope of Pharmaceutics, there are several major concerns that must be addressed before further consideration:

  1. The authors must discuss about Soluplus (Polyvinyl caprolactam–polyvinylacetate-polyethylene glycol-graft-co-polymer) briefly in the introduction, why soluplus and its use in the drug delivery. The authors tried to explain on page 11, lines 478-489, better to move this into introductory parts.
  2. On the Gelatin/mannose preparation and characterization, the authors simply combined the two-sample solution under stirring, what is the bond between gelatin and mannose? In addition, 1h of dialysis is very low to remove unconjugated or unreacted materials which lead to wrong results for the ATR-FTIR or 1H-NMR. The author must do dialysis for a longer time at least 24-48h by exchanging the external buffer within a specified time and then re-do the ATR-FTIR or 1H-NMR. 
  3. On page 4, Micellar preparation and drug encapsulation, the authors must explain how unencapsulated RIF and CUR were removed from the micelle. Simple filtering using 0.45 μm, acetate cellulose filters, is not convincing. 
  4. In this experiment, mannose was used as the ligands to decorate the micelle for the target delivery but what is the importance of gelatin, authors must explain the importance of gelatin in this work. Why not decorate their primary Soluplus polymer with Mannose than mixing with Gel/Man?
  5. The particle size and PDI results of BSA in Table 1 are irrelevant better to remove from table 1.
  6. The authors claimed that RIF and CUR co-loaded micelle had better antioxidant activity than the free drugs, as shown in Tables 5 and 6. However, the comparison was made between RIF and CUR co-loaded micelles and every single free drug. The authors must include the antioxidant effects of combined free drugs (RIF +CUR) to make this type of comparison.  
  7. In figure 4, The authors mentioned that mannose surface decoration did not affect the RIF release from those micellar systems. However, as shown in figure 4, the RIF release was higher in the mannose decorated micelle than in the other. E.g., after 24h of dialysis almost ~100% RIF was released in the mannose surface decorated micelle than the other micelle (~80%). In addition, although the authors mentioned CUR showed a different release behavior, regardless of the presence of mannose, the amount of CUR release looks less affected by mannose surface decoration (after 24 h, CUR release was 56.2% and 51.1% for the micellar systems with and without mannose, respectively) in comparison to the RIF. Needs more explanation or corrections.
  8. Better to write the summary of statistical analysis at the end of materials and methods or before results and discussion. Remove 2.6.3. Statistical analysis and keeps only 2.12.4. Statistical analysis.

Author Response

The authors have tried to synthesize inhalable mannosylated rifampicin-curcumin co-loaded nanomicelles with enhanced in vitro antimicrobial efficacy for an optimized pulmonary tuberculosis therapy and characterized in terms of micellar size, size distribution, and morphology. Moreover, the in vitro nebulization aptitude of formulations and their aerodynamic diameter were evaluated along with their hemolytic potential. Finally, the microbicidal effectiveness of formulations and the micellar accumulation in the lungs has been investigated in vitro by using THP-1 infected (Mtb H37Rv) macrophages and in vivo by using Wistar rats, respectively. The paper's subject is interesting, and the authors have tried performing several in vitro and in vivo experiments to support their experimental findings. While this study falls within the scope of Pharmaceutics, there are several major concerns that must be addressed before further consideration:

  1. The authors must discuss about Soluplus (Polyvinyl caprolactam–polyvinylacetate-polyethylene glycol-graft-co-polymer) briefly in the introduction, why soluplus and its use in the drug delivery. The authors tried to explain on page 11, lines 478-489, better to move this into introductory parts.

We thanks the Reviewer for the suggestion, the description of the Soluplus has been moved from Discussion to Introduction section accordingly.

  1. On the Gelatin/mannose preparation and characterization, the authors simply combined the two-sample solution under stirring, what is the bond between gelatin and mannose? In addition, 1h of dialysis is very low to remove unconjugated or unreacted materials which lead to wrong results for the ATR-FTIR or 1H-NMR. The author must do dialysis for a longer time at least 24-48h by exchanging the external buffer within a specified time and then re-do the ATR-FTIR or 1H-NMR. 

We understand the reviewer point of view. As it was described in the manuscript, it was observed an interaction between gelatin and mannose (for instance the C–O–C asymmetric stretching vibration frequency of mannose at 1136 cm-1) in the ATR-FTIR analysis. This situation was completely different from a physical mixture between gelatin + 5% mannose where the sum of the bands in the IR spectrum of both was observed. To gain further insight in the nature of this interaction, a 1H-NMR study was performed. In this case it was not observed a covalent conjugation due to the absence of resonance signals in the aromatic region of gelatin (formation of an imine or Schiff's base). However, it was observed a molecular interaction between mannose and gelatin as changes associated with proline residues close to δ1H ~ 2 ppm and glutamic acid residues at δ1H ~ 2.3 ppm were detected. In addition, mannose moieties on the surface of the micellar nanocarrier retained their functional ability to interact with mannose-binding lectin, as it was observed in the Concanavalin A assay.

Afterwards, we used the dialysis method to remove the free mannose under magnetic stirring (50 rpm). As it is described in the manuscript, the external medium (distilled water, 1000 mL) was replaced every 15 minutes to enhance the mannose release. The volume of sample inside the dialysis membrane was 25 mL. This method has been previously reported for the mannosylation of solid lipid nanoparticles. In these investigations, the mannose-modified nanoparticles were dialysed against distilled water at room temperature for 30 minutes (Vieira ACC. et al. Artificial Cells, Nanomedicine, and Biotechnology, DOI: 10.1080/21691401.2018.1434186) and 1 h (Costa A. et al. European Journal of Pharmaceutical Sciences 114 (2018) 103–113) in order to remove free mannose.

  1. On page 4, Micellar preparation and drug encapsulation, the authors must explain how unencapsulated RIF and CUR were removed from the micelle. Simple filtering using 0.45 μm, acetate cellulose filters, is not convincing. 

We understand the reviewer concern. As it is well known, polymeric micelles are dynamic colloidal nanocarriers composed of a core-shell structure (a hydrophobic core and a hydrophilic corona) where the amphiphilic biopolymers self-assemble in aqueous upon their critical micellar concentration. According with the physicochemical properties of the encapsulated drug, they could be hosted within the micellar core (hydrophobic drugs) or even between the interface of the micellar corona and the core (amphiphilic drugs) [Owen SC. et al. NanoToday 7, (2012) 53-65; Ghezzi M et al. Journal of Controlled Release 332 (2021) 312–336 https://doi.org/10.1016/j.jconrel.2021.02.031]. In recent years our research group has developed a variety of micellar nannocarriers employing different biopolymers for encapsulation of poorly-water soluble drugs such as paclitaxel, curcumin, doxorubicin, carvedilol, rifampicin, nelfinavir mesilate and nevirapine [Moretton MA. et al. Colloids and Surfaces B: Biointerfaces 123 (2014) 302–310; Wegmann M. et al. Journal of Pharmacy and Pharmacology 69 (2017)544-553; Cagel M. et al. Biomedicine & Pharmacotherapy 95 (2017) 894–903; Grotz E. et al. Journal of Drug Delivery Science and Technology 53 (2019) 101170; Riedel J. et al. Journal of Drug Delivery Science and Technology 62 (2021) 102343]. In the present investigation, upon encapsulation within the micellar dispersions, the un-dissolved drugs were removed by filtration employing cellulose filters (0.45 μm). We used this methodology since no differentiation is required between the encapsulated and the free drug due to low water solubility of hydrophobic drugs for micellar dispersions. Then, the amount of drug in the micelles can be quantified spectrophotometrically or via HPLC analysis after separation of the undissolved drug. On the other hand, a separation method as ultrafiltration spin columns is recommended for hydrophilic drugs encapsulation within polymeric micelles before drug quantification [Ghezzi M et al. Journal of Controlled Release 332 (2021) 312–336 https://doi.org/10.1016/j.jconrel.2021.02.031 ].    

  1. In this experiment, mannose was used as the ligands to decorate the micelle for the target delivery but what is the importance of gelatin, authors must explain the importance of gelatin in this work. Why not decorate their primary Soluplus polymer with Mannose than mixing with Gel/Man?

We understand the reviewer point of view. On the one hand, developments involving the chemical modification of biopolymers with mannose usually require the use of organic solvents as THF and DMSO [D'Addio SM et al. Journal of Controlled Release 168 (2013) 41–49; Freichels H. et al. Bioconjugate Chem. 2012, 23, 1740−1752; Lv S. et al. Adv. Healthcare Mater.2021, 2101651]. Taking into account the pharmaceutical application of our nanoformulation, we aimed to developed mannosylated polymeric micelles employing more biocompatible solvents in a simple and organic solvent free reaction.

On the other hand, gelatine is a natural biopolymer encoded as a GRAS material by the United States Food and Drug Administration (FDA) [United States Food and Drud Administration (FDA). Available online: http://wayback.archive-it.org/7993/20171031062708/https://www.fda.gov/Food/IngredientsPackagingLabeling/GRAS/SCOGS/ucm261307.htm]. It presents a great potential for pharmaceutical dosage forms as it is biocompatible, biodegradable and low-immunogenic. Thereafter, gelatin has been widely employed in pharmaceutical products and novel drug delivery systems. Particularly, gelatin exhibits a great potential for chemical modifications [Elzoghby, A.O. Gelatin-based nanoparticles as drug and gene delivery systems: reviewing three decades of research. J. Control. Release. 2013, 172, 1075-91]. In this framework, we aimed to obtain mannose-modified gelatin employing an organic solvent free technique, as previously reported by Vieira 2018 and Costa 2018 [Vieira ACC. et al. Artificial Cells, Nanomedicine, and Biotechnology, DOI: 10.1080/21691401.2018.1434186; Costa A. et al. European Journal of Pharmaceutical Sciences 114 (2018) 103–113]. One of the advantages of this method is being able to use renewable materials like gelatine and mannose, which are eco-friendly agents.

The particle size and PDI results of BSA in Table 1 are irrelevant better to remove from table 1.

BSA results have been removed according to the Reviewer’ suggestion

  1. The authors claimed that RIF and CUR co-loaded micelle had better antioxidant activity than the free drugs, as shown in Tables 5 and 6. However, the comparison was made between RIF and CUR co-loaded micelles and every single free drug. The authors must include the antioxidant effects of combined free drugs (RIF +CUR) to make this type of comparison.

We understand the reviewer point of view. For a better understanding, we have added the in vitro antioxidant activity of RIF + CUR dispersion in both Tables (5 and 6). Further we have added these results to the discussion in Section 3.5. In vitro antioxidant capacity

  1. In figure 4, The authors mentioned that mannose surface decoration did not affect the RIF release from those micellar systems. However, as shown in figure 4, the RIF release was higher in the mannose decorated micelle than in the other. E.g., after 24h of dialysis almost ~100% RIF was released in the mannose surface decorated micelle than the other micelle (~80%). In addition, although the authors mentioned CUR showed a different release behavior, regardless of the presence of mannose, the amount of CUR release looks less affected by mannose surface decoration (after 24 h, CUR release was 56.2% and 51.1% for the micellar systems with and without mannose, respectively) in comparison to the RIF. Needs more explanation or corrections.

We apologize for the incorrect explanation of the release of both bioactives, the text has been modified accordingly.

  1. Better to write the summary of statistical analysis at the end of materials and methods or before results and discussion. Remove 2.6.3. Statistical analysis and keeps only 2.12.4. Statistical analysis.

We thank the Reviewer for the remark, Statistical analyses have been merged in one paragraph, accordingly

Round 2

Reviewer 2 Report

Q1: Authors did not answer to my question. I think this is a mistake and that is the answer for some of the other reviewers. 

Q2: Maybe I missed it but I cannot find in the text that the experiments are done with the freeze-dried sample. If this is the case, please note it. 

Q3: If the authors are reporting a targeted system, it has to be compared to the non-targeted material. Otherwise, it is impossible to know if this new material is better than the regular one. If the authors have already reported the non-targeted material, I assume they will be able to compare, at least, accumulation and toxicity. 

Author Response

Reviewer 2

Q1: Authors did not answer to my question. I think this is a mistake and that is the answer for some of the other reviewers.

We are really sorry that the Reviewer does not consider our answer complete to the question but we have detailed in the manuscript what it is observed. Besides, the main goal of our nanoformulation was to demonstrate as it was stated in the Introduction “Hence, the present investigation was aimed at expanding the potential of this inhalable nanotechnological platform for an active drug targeting to the Mtb-infected macrophages.”

We described in the manuscript that there was a molecular interaction between mannose and gelatin. This interaction was confirmed by ATR-FTIR and 1HNMR. Further, surface mannosylation of the micelles was confirmed by a lectin-binding assay. Then, the mannose coating demonstrated to be useful even under dilution for the in vitro microbicidal assays (samples were diluted from a RIF concentration of 10 mg/mL to 5 µg/mL). Then this nanoformulation could be an excellent platform for a respirable anti-TB therapy where the dilution of the micellar system upon administration would be much lower than after an oral or parenteral administration.

Finally, we mentioned in the Conclusions section that “Further studies employing an in vivo TB-infection model will be performed to investigate the anti-TB performance of this nanoformulation to optimize TB pulmonary therapy.”

Q2: Maybe I missed it but I cannot find in the text that the experiments are done with the freeze-dried sample. If this is the case, please note it.

We thank the reviewer for the remark. We have improved the Material and Methods Section and we have fully detailed in every assay that freeze-dried micelles were employed.

Q3: If the authors are reporting a targeted system, it has to be compared to the non-targeted material. Otherwise, it is impossible to know if this new material is better than the regular one. If the authors have already reported the non-targeted material, I assume they will be able to compare, at least, accumulation and toxicity. 

We understand the Reviewer point of view but in this particular case, the main objective of the investigation was the development of a micellar system for active targeting to macrophages (cellular reservoir of M. tuberculosis), while an anatomical targeting to the lungs was not the objective of the assay. Instead, the main objective of the in vitro lung accumulation assay was to confirm if the mannose-coated micelles could be accumulated in the lungs after its intratracheal administration for a respirable formulation. Taking into account our previous findings, which have demonstrated that Soluplus micelles could be accumulated in these organs over 24 h. Hence, the incorporation of the mannose-modified gelatin to the micellar system could alter the micelle’s fate after intratracheal administration. Then, we demonstrated that mannose-coated micelles were also capable of effectively accumulating in the lungs over 24 h. Otherwise, they won’t be able to target the alveolar macrophages.

Moreover, the significant enhancement of the in vitro microbicidal efficacy of the drug-loaded mannose-coated micelles versus their mannose-free counterparts denotes their potential as an inhalable system for macrophage targeting.

Round 3

Reviewer 2 Report

Q1: My first question was: "I understand from the protocols that the interaction between mannose and gelatine is rather electrostatic than covalent. If so, the authors should demonstrate that such interaction is not going to weaken over time, since that could lead to non-targeted micelles.". Hence, I can't see how the text provided by the authors is answering my question, despite the authors claiming so. 

Q2: OK

Q3: I am afraid I can't agree with the authors. My question is: if they don't compare with the non-targeted material, how can they be so sure that there is an advancement in this work? In other words: How can they be sure that the mannose coating is providing an advantage if they don't do the same experiment with non-targeted materials? Otherwise, it is impossible to discern whether the new formulation is beneficial or it just produces a more complicated system.  

Author Response

Q1: My first question was: "I understand from the protocols that the interaction between mannose and gelatine is rather electrostatic than covalent. If so, the authors should demonstrate that such interaction is not going to weaken over time, since that could lead to non-targeted micelles.". Hence, I can't see how the text provided by the authors is answering my question, despite the authors claiming so.

We agree with the reviewer. In the 1H-NMR analysis we can indirectly observe the presence of weak interactions (primarily hydrogen bonds) as evidenced by changes in some of the hydrophobic residues of the amino acids. Some of these interactions likely cause a change at the supramolecular level in the conformational structure of gelatin as it is evident in the mannose-modified gelatin sample. However, these changes are not observed in either FTIR or 1H-NMR spectra after the analysis of the physical mixture of gelatin and mannose. Hence promising results were observed in the in vitro efficacy assays for the mannose-coated micelles. Then, further in vivo assays would be required to further explore the fate of this mannosylated nanocarrier after pulmonary administration in an Mtb infection model. This conclusion has been stated in the Conclusion Section as follows: “Further studies employing an in vivo TB-infection model will be performed to investigate the anti-TB performance of this nanoformulation to optimize TB pulmonary therapy”.

Q3: I am afraid I can't agree with the authors. My question is: if they don't compare with the non-targeted material, how can they be so sure that there is an advancement in this work? In other words: How can they be sure that the mannose coating is providing an advantage if they don't do the same experiment with non-targeted materials? Otherwise, it is impossible to discern whether the new formulation is beneficial or it just produces a more complicated system. 

We agree with the reviewer. It would be crucial to compare the mannose-free and mannose-coated systems if the goal of the investigation is an anatomical targeting. Some examples are nanotechnological developments for targeting to the central nervous system after intranasal administration [Chiappetta DA et al. 2013 Nanomedicine 8  https://doi.org/10.2217/nnm.12.104] or targeting to solid tumours after parenteral administration [Danhier F. et al. 2010, Journal of Controlled Released 148, 135-146]. In this case, the nanaocarrier will be distributed in the body after their administration and they will be selectively accumulated in a specific organ/tumour.

However, our main objective was a cellular targeting to the macrophages after the administration of this mannose-coated system directly to the respiratory system. In this case, micelles should remain accumulated in the lungs otherwise the targeting to macrophages could not be observed. This is the main reason why we performed the in vivo assays with the freeze-dried mannose-coated micelles. This assay was performed considering our previous experience with the uncoated counterparts. As it can be see in the Figure from Grotz et al 2019, “The biodistribution profile after 1 h of administration (Fig. 6) showed that 84% of the radiolabel micelles remained in the body with a 16% of renal elimination. Particularly, 76% of the remained fraction in the body showed pulmonary accumulation” “After a 24 h of administration, 65% of the 99mTc-micelles remained in the body denoting renal elimination. In this case, most of the micelles (91%) were accumulated in the lungs with no evidence of micelle presence in the trachea” [Grotz et at. 2019 Journal of Drug Delivery Science and Technology 53, 101170]. Then, the next step was to expand the potential of this micellar nanocarrier by the coencapsulation of rifampicin and curcumin along with the mannose coating of the polimeric micelles. We hope that we could clarify the main objective of our assay as it was requested by the reviewer.

Round 4

Reviewer 2 Report

Q1: I will try to clarify my question: Have the authors studied the long-term stability of system? In other words, given that the final formulation is going to be the freeze-dried one, how long can it be stored in that state before loosing its properties (i.e, everything bonded electrostatically so the system retains its targeting ability)?

Q3: The authors' answer is reinforcing what I say. According to their previous article "In this case, most of the micelles (91%) were accumulated in the lungs with no evidence of micelle presence in the trachea". Similarly, in their statement they say "In this case, micelles should remain accumulated in the lungs otherwise the targeting to macrophages could not be observed. This is the main reason why we performed the in vivo assays with the freeze-dried mannose-coated micelles." This directly means that without the coating there would not be targeting. However, attending to what they have said both in the article and in the answer to my question, both nanoparticles accumulate in the lungs. Hence, my question is, why don't you compare with the non-targeted one so you can establish whether implementing the mannose coating is worth it or not?